# A Reduction-based Framework for Sequential Decision Making with Delayed Feedback

**Yunchang Yang**[1][*]  **Han Zhong**[1][*]  **Tianhao Wu**[2][*]  **Bin Liu**[3]  **Liwei Wang**[1,4]  **Simon S. Du**[5]
[1]Center for Data Science, Peking University
[2]University of California, Berkeley    [3]Zhejiang Lab
[4]National Key Laboratory of General Artificial Intelligence,
School of Intelligence Science and Technology, Peking University
[5]University of Washington

## Abstract

We study stochastic delayed feedback in general sequential decision-making problems, which include bandits, single-agent Markov decision processes (MDPs), and Markov games (MGs). We propose a novel reduction-based framework, which turns any multi-batched algorithm for sequential decision making with instantaneous feedback into a sample-efficient algorithm that can handle stochastic delays in sequential decision-making problems. By plugging different multi-batched algorithms into our framework, we provide several examples demonstrating that our framework not only matches or improves existing results for bandits, tabular MDPs, and tabular MGs, but also provides the first line of studies on delays in sequential decision making with function approximation. In summary, we provide a complete set of sharp results for single-agent and multi-agent sequential decision-making problems with delayed feedback.

## 1   Introduction

Delayed feedback is a common and fundamental problem in sequential decision making (Sutton and Barto, 2018; Lattimore and Szepesvári, 2020; Zhang et al., 2021a). Taking the recommendation system as an example, delayed feedback is an inherent part of this problem. Specifically, the learner adjusts her recommendation after receiving users' feedback, which may take a random delay after the recommendation is issued. More examples include but are not limited to robotics (Mahmood et al., 2018) and video steaming (Changuel et al., 2012). Furthermore, the delayed feedback issue is more serious in multi-agent systems (Chen et al., 2020) since observing the actions of other agents' may happen with a varying delay.

Although handling delayed feedback is a prominent challenge in practice, the theoretical understanding of delays in sequential decision making is limited. Even assuming the delays are stochastic (delays are sampled from a fixed distribution), the nearly minimax optimal result is only obtained by works on multi-armed bandits and contextual bandits (Agarwal and Duchi, 2011; Dudik et al., 2011; Joulani et al., 2013; Vernade et al., 2020, 2017; Gael et al., 2020). For more complex problems such as linear bandits (Lancewicki et al., 2021) and tabular reinforcement learning (RL) (Howson et al., 2022), the results are suboptimal. Meanwhile, to the best of our knowledge, there are no theoretical guarantees in RL with function approximation and multi-agent RL settings. Hence, we focus on stochastic delays and aim to answer the following two questions in this work:

1. Can we provide sharper regret bounds for basic models such as linear bandits and tabular RL?

---

[*]Equal contribution.  Correspondence to Yunchang Yang ⟨yangyc@pku.edu.cn⟩ and Han Zhong ⟨hanzhong@stu.pku.edu.cn⟩.

37th Conference on Neural Information Processing Systems (NeurIPS 2023).

Table 1: Comparison of bounds for MSDM with delayed feedback. We denote by $K$ the number of rounds the agent plays (episodes in RL), $A$ and $B$ the number of actions (or arms), $A_{\max} = \max_i A_i$, $S$ the number of states, $H$ the length of an episode, $d$ the linear feature dimension, and $\dim_E(\mathcal{F}, 1/K)$ the eluder dimension of general function class $\mathcal{F}$, $d_\tau(q)$ is the quantile function of the random variable $\tau$ and $n$ is the number of players.

| Setting | Previous Result | This Work |
|---|---|---|
| Multi-armed bandit | $O(\frac{A\log(K)}{q\min_{i\neq\star}\Delta_i} + d_\tau(q)\log K)$ (Lancewicki et al., 2021) | $O\left(\frac{A\log K}{q\min_{i\neq\star}\Delta_i} + \frac{\log^2 K}{q} + d_\tau(q)\log K\right)$ |
| | | $O\left(\frac{A\log K}{\min_{i\neq\star}\Delta_i} + \mathbb{E}[\tau]\log K\right)$ |
| Linear bandit | $\tilde{O}(d\sqrt{K} + d^{3/2}\mathbb{E}[\tau])$ (Howson et al., 2021) $\tilde{O}(d\sqrt{K} + \mathbb{E}[\tau])$ (Vakili et al., 2023), concurrent work | $\tilde{O}\left(\frac{1}{q}d\sqrt{K} + d_\tau(q)\right)$ |
| | | $\tilde{O}(d\sqrt{K} + \mathbb{E}[\tau])$ |
| Tabular MDP | $\tilde{O}(\sqrt{H^3SAK} + H^2SA\mathbb{E}[\tau]\log K)$ (Howson et al., 2022) | $\tilde{O}\left(\frac{1}{q}\sqrt{SAH^3K} + H(H+\log\log K)d_\tau(q)\right)$ |
| | | $\tilde{O}\left(\sqrt{SAH^3K} + H(H+\log\log K)\mathbb{E}[\tau]\right)$ |
| Linear MDP | — | $\tilde{O}(\frac{1}{q}\sqrt{d^3H^4K} + dH^2d_\tau(q)\log K)$ |
| | | $\tilde{O}(\sqrt{d^3H^4K} + dH^2\mathbb{E}[\tau])$ |
| RL with general function approximation | — | $\tilde{O}(\frac{1}{q}\sqrt{\dim_E^2(\mathcal{F},1/K)H^4K} + H^2\dim_E(\mathcal{F},1/K)d_\tau(q))$ |
| | | $\tilde{O}\left(\sqrt{\dim_E^2(\mathcal{F},1/K)H^4K} + H^2\dim_E(\mathcal{F},1/K)\mathbb{E}[\tau]\right)$ |
| Tabular two-player zero-sum MG | — | $\tilde{O}(\frac{1}{q}\left(\sqrt{H^3SABK} + H^3S^2AB\right) + H^2SABd_\tau(q)\log K)$ |
| | | $\tilde{O}(\sqrt{H^3SABK} + H^3S^2AB + H^2SAB\mathbb{E}[\tau]\log K)$. |
| Linear two-player zero-sum MG | — | $\tilde{O}\left(\frac{1}{q}\sqrt{d^3H^4K} + dH^2d_\tau(q)\log K\right)$ |
| | | $\tilde{O}\left(\sqrt{d^3H^4K} + dH^2\mathbb{E}[\tau]\log K\right)$ |
| Tabular multi-player general-sum MG | $\tilde{O}\left(H^3\sqrt{SA_{\max}} + H^3\sqrt{S\mathbb{E}[\tau]K}\right)$ (Zhang et al., 2022a) | $\tilde{O}(\frac{1}{q}\sqrt{H^5SA_{\max}K} + H^2nSd_\tau(q)\log K)$ |
| | | $\tilde{O}(\sqrt{H^5SA_{\max}K} + H^2nS\mathbb{E}[\tau]\log K)$ |

## 2. Can we handle delays in (multi-agent) sequential decision making in the context of function approximation?

We answer these two questions affirmatively by proposing a reduction-based framework for stochastic delays in both single-agent and multi-agent sequential decision making. Our contributions are summarized below.

**Our Contributions.** Our main contribution is proposing a new reduction-based framework for both single-agent and multi-agent sequential decision making with stochastic delayed feedback. The proposed framework can convert any multi-batched algorithm (cf. Section 3) for sequential decision making with instantaneous feedback to a provably efficient algorithm that is capable of handling stochastic delayed feedback in sequential decision making. Unlike previous works which require case by case algorithm design and analysis, we provide a unified framework to solve a wide range of problems all together.

Based on this framework, we obtain a complete set of results for single-agent and multi-agent sequential decision making with delayed feedback. In specific, our contributions can be summarized as follows:

- **New framework:** We propose a generic algorithm that can be integrated with any multi-batched algorithm in a black-box fashion. Meanwhile, we provide a unified theoretical analysis for the proposed generic algorithm.

- **Improved results and new results:** By applying our framework to different settings, we obtain state-of-the-art regret bounds for multi-armed bandits, and derive sharper results for linear bandits and tabular RL, which significantly improve existing results (Lancewicki et al., 2021; Howson et al., 2022). We also provide the first result for single-agent RL with delayed feedback in the context of linear or even general function approximation;

- **New algorithms:** We show that delayed feedback in Markov games (MGs) can also be handled by our framework. By plugging several new proposed multi-batched algorithms for MGs into our framework, we not only improve recent results on tabular MGs (Zhang et al., 2022a), but

also present a new result for linear MGs. As a byproduct, we provide the first line of study on multi-batched algorithms for MGs, which might be of independent interest.

Our results and comparisons with existing works are summarized in Table 1.

## 1.1 Related Works

**Bandit/MDP with delayed feedback.** Stochastically delayed feedback has been studied a lot in the multi-armed bandit and contextual bandit problems (Agarwal and Duchi, 2011; Dudik et al., 2011; Joulani et al., 2013; Vernade et al., 2020, 2017; Gael et al., 2020; Huang et al., 2023). A recent work (Lancewicki et al., 2021) shows that under stochastic delay, elimination algorithm performs better than UCB algorithm, and gives tight upper and lower bounds for multi-armed bandit setting. Howson et al. (2021) solves the linear bandit setting by adapting LinUCB algorithm. The concurrent work of Vakili et al. (2023) provides a bound for the kernel bandit setting with delayed feedback. Our work does not consider the kernel setting, but we conjecture that our framework can handle this problem by designing a multi-batched algorithm for kernel bandits.

In RL, despite practical importance, there is limited literature on stochastical delay in RL. Lancewicki et al. (2022) and Jin et al. (2022b) considered adversarial delays. Their regret depends on the sum of the delays, the number of states and the number of steps per episode. However, when reducing their results to stochastic delay, the bounds will be too loose. Howson et al. (2022) considered stochastic delay in tabular MDP, but their bound is not tight. As far as we know, there is no work considering linear MDP and general function approximation with delayed feedback.

Another line of work considers adversarially delayed feedback (Lancewicki et al., 2022; Jin et al., 2022b; Quanrud and Khashabi, 2015; Thune et al., 2019; Bistritz et al., 2019; Zimmert and Seldin, 2020; Ito et al., 2020; Gyorgy and Joulani, 2021; Van Der Hoeven and Cesa-Bianchi, 2022). In the adversarial setting the state-of-the-art result is of form $\sqrt{K + D}$, where $K$ is the number of the episodes and $D$ is the total delay. Note that when adapting their result to the stochastic setting, the bound becomes $\sqrt{K + K\tau}$ where $\tau$ is the expectation of delay, while in stochastic delay setting the upper bound can be $\sqrt{K} + \tau$. Therefore, a direct reduction from adversarial delay setting to stochastic delay setting will result in a suboptimal regret bound.

**Low switching cost algorithm.** Bandit problem with low switching cost has been widely studied in past decades. Cesa-Bianchi et al. (2013) showed an $\tilde{\Theta}(K^{\frac{2}{3}})$ regret bound under adaptive adversaries and bounded memories. Perchet et al. (2016) proved a regret bound of $\tilde{\Theta}(K^{\frac{1}{1-2^{1-M}}})$ for the two-armed bandit problem within $M$ batches, and later Gao et al. (2019) extended their result to the general $A$-armed case. In RL, by doubling updates, the global switching cost is $O(SAH \log_2(K))$ while keeping the regret $\tilde{O}(\sqrt{SAKH^3})$ (Azar et al., 2017). Recently Zhang et al. (2022b) proposed a policy elimination algorithm that achieves $\tilde{O}(\sqrt{SAKH^3})$ regret and $O(H + \log_2 \log_2(K))$ switching cost. Besides, Gao et al. (2021) generalized the problem to linear MDPs, and established a regret bound of $\tilde{O}(\sqrt{d^3H^4K})$ with $O(dH \log(K))$ global switching cost. Recent work Qiao et al. (2022) achieved $O(HSA \log_2 \log_2(K))$ switching cost and $\tilde{O}(\text{poly}(S, A, H)\sqrt{K})$ regret with a computational inefficient algorithm. And Kong et al. (2021) consider MDP with general function approximation settings.

**Markov Games.** A standard framework to capture multi-agent RL is Markov games (also known as stochastic games) (Shapley, 1953). For two-player zero-sum Markov games, the goal is to solve for the Nash equilibrium (NE) (Nash, 1951). Recently, many works study this problem under the generative model (Zhang et al., 2020; Li et al., 2022), offline setting (Cui and Du, 2022b,a; Zhong et al., 2022; Xiong et al., 2022a; Yan et al., 2022), and online setting (Bai and Jin, 2020; Bai et al., 2020; Liu et al., 2021; Tian et al., 2021; Xie et al., 2020; Chen et al., 2021; Jin et al., 2022a; Xiong et al., 2022b; Huang et al., 2021). Our work is mostly related to Liu et al. (2021) and Xie et al. (2020). Specifically, Liu et al. (2021) studies online tabular two-player zero-sum MGs and proposes an algorithm with $O(\sqrt{SABH^3K})$ regret. Xie et al. (2020) focused on online linear MGs provide an algorithm that enjoys $O(\sqrt{d^3H^4K})$ regret. We also remark that Jin et al. (2022a); Xiong et al. (2022b); Huang et al. (2021) establish sharper regret bound for linear MGs but the algorithms therein are computationally inefficient. There is also a line of works (Liu et al., 2021; Jin et al., 2021;

Song et al., 2021; Zhong et al., 2021; Mao and Başar, 2022; Cui and Du, 2022b; Jin et al., 2022c; Daskalakis et al., 2022) studying multi-player general-sum MGs. Among these works, Jin et al. (2021); Song et al. (2021); Mao and Başar (2022) use V-learning algorithms to solve coarse correlated equilibrium (CCE). But all these algorithms can not handle delayed feedback nor are multi-batched algorithms that can be plugged into our framework. To this end, we present the first line of studies on multi-batched algorithms for MGs and provide a complete set of results for MGs with delayed feedback.

A very recent but independent work (Zhang et al., 2022a) also studies multi-agent reinforcement learning with delays. They assume that the reward is delayed, and aim to solve for coarse correlated equilibrium (CCE) in general-sum Markov games. Our work has several differences from Zhang et al. (2022a). First, they study the setting where the total delay is upper bounded by a certain threshold, while we focus on the setup with stochastic delay. Besides, their method seems only applicable for reward delay, while we allow the whole trajectory feedback to be delayed. Second, in the stochastic delay setting the delay-dependent term in their bound is worse than ours. Specifically, their bound scales as $O(H^3\sqrt{S\mathbb{E}[\tau]K})$, while ours is $O(H^2nS\mathbb{E}[\tau])$ where $n$ is the number of players. Finally, it seems that their results cannot be extended to the multi-agent RL with function approximation.

## 2 Preliminary

The objective of this section is to provide a unified view of the settings considered in this paper, i.e., bandits, Markov Decision Processes (MDPs) and multi-agent Markov Games. We consider a general framework for interactive decision making, which we refer to as Multi-agent Sequential Decision Making (MSDM).

**Notations** We use $\Delta(\cdot)$ to represent the set of all probability distributions on a set. For $n \in \mathbb{N}_+$, we denote $[n] = \{1, 2, \ldots, n\}$. We use $O(\cdot), \Theta(\cdot), \Omega(\cdot)$ to denote the big-O, big-Theta, big-Omega notations. We use $\widetilde{O}(\cdot)$ to hide logarithmic factors.

**Multi-agent Sequential Decision Making** A Multi-agent Sequential Decision Making problem can be represented as a tuple $M = (n, \mathcal{S}, \{\mathcal{A}_i\}_{i=1}^n, H, \{p_h\}_{h=1}^H, s_1, \{r_h\}_{h=1}^H)$, where $n$ is the number of agents, $\mathcal{S}$ is the state space, $\mathcal{A}_i$ is the action space of player $i$, $H$ is the length of each episode and $s_1$ is the initial state. We define $\mathcal{A} = \bigotimes_i \mathcal{A}_i$ as the whole action space.

At each stage $h$, for every player $i$, every state-action pair $(s, a_1, ..., a_n)$ is characterized by a reward distribution with mean $r_{i,h}(s, a_1, ..., a_n)$ and support in $[0, 1]$, and a transition distribution $p_h(\cdot|s, a)$ over next states. We denote by $S = |\mathcal{S}|$ and $A_i = |\mathcal{A}_i|$.

The protocol proceeds in several episodes. At the beginning of each episode, the environment starts with initial state $s_1$. At each step $h \in [1, H]$, the agents observe the current state $s_h$, and take their actions $(a_1, ..., a_n)$ respectively. Then the environment returns a reward signal, and transits to next state $s_{h+1}$.

A (random) policy $\pi_i$ of the $i^{\text{th}}$ player is a set of $H$ maps $\pi_i := \{\pi_{i,h} : \Omega \times \mathcal{S} \to \mathcal{A}_i\}_{h \in [H]}$, where each $\pi_{i,h}$ maps a random sample $\omega$ from a probability space $\Omega$ and state $s$ to an action in $\mathcal{A}_i$. A joint (potentially correlated) policy is a set of policies $\{\pi_i\}_{i=1}^m$, where the same random sample $\omega$ is shared among all agents, which we denote as $\pi = \pi_1 \odot \pi_2 \odot \ldots \odot \pi_m$. We also denote $\pi_{-i}$ as the joint policy excluding the $i^{\text{th}}$ player. For each stage $h \in [H]$ and any state-action pair $(s, a) \in \mathcal{S} \times \mathcal{A}$, the value function and Q-function of a policy $\pi$ are defined as:

$$Q_{i,h}^\pi(s, a) = \mathbb{E}\left[ \sum_{h'=h}^H r_{i,h'} \,\Big|\, s_h = s, a_h = a, \pi \right], \quad V_{i,h}^\pi(s) = \mathbb{E}\left[ \sum_{h'=h}^H r_{i,h'} \,\Big|\, s_h = s, \pi \right].$$

For each policy $\pi$, we define $V_{H+1}^\pi(s) = 0$ and $Q_{H+1}^\pi(s, a) = 0$ for all $s \in \mathcal{S}, a \in \mathcal{A}$. Sometimes we omit the index $i$ when it is clear from the context.

For the single agent case ($n = 1$), the goal of the learner is to maximize the total reward it receives. There exists an optimal policy $\pi^\star$ such that $Q_h^\star(s, a) = Q_h^{\pi^\star}(s, a) = \max_\pi Q_h^\pi(s, a)$ satisfies the optimal Bellman equations

$$Q_h^\star(s, a) = r_h(s, a) + \mathbb{E}_{s' \sim p_h(s, a)}[V_{h+1}^\star(s')], \quad V_h^\star(s) = \max_{a \in \mathcal{A}}\{Q_h^\star(s, a)\}, \quad \forall (s, a) \in \mathcal{S} \times \mathcal{A}.$$

Then the optimal policy is the greedy policy $\pi_h^\star(s) = \arg\max_{a \in \mathcal{A}}\{Q_h^\star(s,a)\}$. And we evaluate the performance of the learner through the notion of single-player regret, which is defined as

$$\text{Regret}(K) = \sum_{k=1}^{K}\left(V_1^\star - V_1^{\pi_k}\right)(s_1).\tag{1}$$

In the case of multi-player general-sum MGs, for any policy $\pi_{-i}$, the best response of the $i^{\text{th}}$ player is defined as a policy of the $i^{\text{th}}$ player which is independent of the randomness in $\pi_{-i}$, and achieves the highest value for herself conditioned on all other players deploying $\pi_{-i}$. In symbol, the best response is the maximizer of $\max_{\pi_i'} V_{i,1}^{\pi_i' \times \pi_{-i}}(s_1)$ whose value we also denote as $V_{i,1}^{\dagger, \pi_{-i}}(s_1)$ for simplicity.

For general-sum MGs, we aim to learn the Coarse Correlated Equilibrium (CCE), which is defined as a joint (potentially correlated) policy where no player can increase her value by playing a different independent strategy. In symbol,

**Definition 1** (Coarse Correlated Equilibrium). *A joint policy $\pi$ is a CCE if $\max_{i \in [m]}(V_{i,1}^{\dagger, \pi_{-i}} - V_{i,1}^{\pi})(s_1) = 0$. A joint policy $\pi$ is a $\epsilon$-approximate CCE if $\max_{i \in [m]}(V_{i,1}^{\dagger, \pi_{-i}} - V_{i,1}^{\pi})(s_1) \le \epsilon$.*

For any algorithm that outputs policy $\hat{\pi}^k$ at episode $k$, we measure the performance of the algorithm using the following notion of regret:

$$\text{Regret}(K) = \sum_{k=1}^{K} \max_{j}\left(V_{j,h}^{\dagger, \hat{\pi}_{-j,h}^k} - V_{j,h}^{\hat{\pi}_h^k}\right)(s_1).\tag{2}$$

One special case of multi-player general-sum MGs is two-player zero-sum MGs, in which there are only two players and the reward satisfies $r_{2,h} = -r_{1,h}$. In zero-sum MGs, there exist policies that are the best responses to each other. We call these optimal strategies the Nash equilibrium of the Markov game, which satisfies the following minimax equation:

$$\sup_{\mu} \inf_{\nu} V_h^{\mu,\nu}(s) = V_h^{\mu^\star, \nu^\star}(s) = \inf_{\nu} \sup_{\mu} V_h^{\mu,\nu}(s).$$

We measure the suboptimality of any pair of general policies $(\hat{\mu}, \hat{\nu})$ using the gap between their performance and the performance of the optimal strategy (i.e., Nash equilibrium) when playing against the best responses respectively:

$$V_1^{\dagger, \hat{\nu}}(s_1) - V_1^{\hat{\mu}, \dagger}(s_1) = \left[V_1^{\dagger, \hat{\nu}}(s_1) - V_1^\star(s_1)\right] + \left[V_1^\star(s_1) - V_1^{\hat{\mu}, \dagger}(s_1)\right].$$

Let $(\mu^k, \nu^k)$ denote the policies deployed by the algorithm in the $k^{\text{th}}$ episode. The learning goal is to minimize the two-player regret of $K$ episodes, defined as

$$\text{Regret}(K) = \sum_{k=1}^{K}\left(V_1^{\dagger, \nu^k} - V_1^{\mu^k, \dagger}\right)(s_1).\tag{3}$$

One can see that (3) is a natural reduction of (2) from general-sum games to zero-sum games.

**MSDM with Delayed Feedback**    We focus on the case where there is delay between playing an episode and observing the sequence of states, actions and rewards sampled by the agent; we refer to this sequence as feedback throughout this paper.

We assume that the delays are stochastic, i.e. we introduce a random variable for the delay between playing the $k$-th episode and observing the feedback of the $k$-th episode, which we denote by $\tau_k$. We assume the delays are i.i.d:

**Assumption 1.** *The delays $\{\tau_k\}_{k=1}^{k}$ are positive, independent and identically distributed random variables: $\tau_k \overset{i.i.d.}{\sim} f_\tau(\cdot)$. We denote the expectation of $\tau$ as $\mathbb{E}[\tau]$ (which may be infinite), and denote the quantile function of $\tau$ as $d_\tau(q)$, i.e.*

$$d_\tau(q) = \min\left\{\gamma \in \mathbb{N} \mid \Pr[\tau \le \gamma] \ge q\right\}.\tag{4}$$

When delay exists, the feedback associated with an episode does not return immediately. Instead, the feedback of the $k$-th episode cannot be observed until the $k + \tau_k$-th episode ends.

Sometimes in order to control the scale and distribution of the delay, we make the assumption that the delays are $(v, b)$ subexponential random variables. This is a standard assumption used in previous works Howson et al. (2021), though it is not always necessary in our paper.

**Assumption 2.** *The delays are non-negative, independent, and identically distributed $(v, b)$- subexponential random variables. That is, their moment generating function satisfies the following inequality*

$$\mathbb{E}\left[\exp\left(\gamma\left(\tau_t - \mathbb{E}\left[\tau_t\right]\right)\right)\right] \leq \exp\left(\frac{1}{2}v^2\gamma^2\right) \tag{5}$$

*for some $v$ and $b$, and all $|\gamma| \leq 1/b$.*

## 3    Multi-batched Algorithm for MSDMs

In this section we mainly discuss multi-batched algorithms for MSDMs. This section aims to provide a formal definition of multi-batched algorithm, and give some examples of multi-batched algorithm for the ease of understanding.

Formally, a multi-batched algorithm consists of several batches. At the beginning of the $m^{\text{th}}$ batch, the algorithm outputs a policy sequence $\pi^m = (\pi_1^m, \pi_2^m, ..., \pi_{l^m}^m)$ for this batch using data collected in previous batches, along with a stopping criteria $SC$, which is a mapping from a dataset to a boolean value. Then the agent executes the policy sequence $\pi^m$ for several episodes and collects feedback to the current dataset $\mathcal{D}$, until the stopping criteria are satisfied (i.e. $SC(\mathcal{D}) = 1$). If the length of the $m^{\text{th}}$ batch exceeds $l_m$, then we simply run the policy sequence cyclically, which means in the $k^{\text{th}}$ episode we run policy $\pi_{k \bmod l^m}^m$. Algorithm 1 describes a protocol of multi-batched algorithm.

---

**Algorithm 1** Protocol of Multi-batched Algorithm

---

1: Initialize dataset $\mathcal{D}^0 = \emptyset$
2: **for** batch $m = 1, ..., M$ **do**
3:     $\mathcal{D} = \emptyset$
4:     $k = 0$
5:     Calculate a policy sequence $\pi^m = (\pi_1^m, \pi_2^m, ..., \pi_{l^m}^m)$ using previous data $\mathcal{D}^{m-1}$, and a stopping criteria $SC$
6:     **while** $SC(\mathcal{D}) = 0$ **do**
7:         $k \leftarrow k + 1$
8:         In episode $k$, execute $\pi_{k \bmod l^m}^m$ and collect trajectory feedback $o^k$
9:         $\mathcal{D} = \mathcal{D} \cup \{o^k\}$
10:    **end while**
11:    Update dataset $\mathcal{D}^m = \mathcal{D}^{m-1} \cup \mathcal{D}$
12: **end for**
13: **Output:** A policy sequence $\{\pi^m\}$

---

Note that the major requirement is that the policy to be executed during the batch should be calculated at the beginning of the batch. Therefore, algorithms that calculate the policy at the beginning of each episode do not satisfy this requirement. For example, UCB algorithms in bandits (Lattimore and Szepesvári, 2020) and UCBVI (Azar et al., 2017) are not multi-batched algorithms.

**Relation with low-switching cost algorithm**    Our definition of multi-batched algorithm is similar to that of low-switching cost algorithm, but is more general.

For any algorithm, the switching cost is the number of policy changes in the running of the algorithm in $K$ episodes, namely:

$$N_{\text{switch}} \triangleq \sum_{k=1}^{K-1} \mathbb{I}\left\{\pi_k \neq \pi_{k+1}\right\}.$$

Most algorithms with $N_{\text{switch}}$ low-switching cost are multi-batched algorithms with $N_{\text{switch}}$ batches. For example, the policy elimination algorithm proposed in Zhang et al. (2022b) is a multi-batched

algorithm with $2H + \log \log K$ batches. And the algorithm for linear MDP with low global switching cost in Gao et al. (2021) is a multi-batched algorithm with $O(dH \log K)$ batches. On the other hand, the difference between batches and switching cost is that we do not require the policy to be the same during a batch. Therefore, a multi-batched algorithm with $N$ batches may not be an algorithm with $N$ switching cost. For example, the Phase Elimination Algorithm in linear bandit with finite arms setting (Lattimore and Szepesvári, 2020) is a multi-batched algorithm with $O(\log K)$ batches, but the switching cost is $O(A \log K)$ which is much larger.

## 4 A Framework for Sequential Decision Making with Delayed Feedback

In this section, we introduce our framework for sequential decision making problems with stochastic delayed feedback. We first describe our framework, which converts any multi-batched algorithm to an algorithm for MSDM with delayed feedback. Then we provide theoretical guarantees for the performance of such algorithms.

Before formally introducing our framework, we first give an intuitive explanation of why multi-batched algorithms work for delayed settings. Recall that Lancewicki et al. (2021) gives a lower bound showing that UCB-type algorithms are suboptimal in multi-armed bandits with stochastic delay (see Theorem 1 in Lancewicki et al. (2021)). The main idea of the lower bound is that UCB algorithm will keep pulling a suboptimal arm until the delayed feedback arrives and the policy is updated. Therefore, if the policy updates too frequently, then the algorithm has to spend more time waiting for the update, and the effect of delay will be greater. This observation motivates us to consider multi-batched algorithms, which have a low frequency of policy updating.

For any sequential decision making problem, assume that we have a multi-batched algorithm $\mathcal{ALG}$ for the problem without delay. Then in the existence of delay, we can still run $\mathcal{ALG}$. The difference is, in each batch of the algorithm, we run some extra steps until we observe enough feedback so that the stopping criteria are satisfied. For example, if the delay $\tau$ is a constant and the stopping criteria is that the policy $\pi_k$ is executed for a pre-defined number of times $t_k$, then we only need to run $\pi_k$ for $t_k + \tau$ times. After the extra steps are finished, we can collect $t_k$ delayed observations, which meet the original stopping criteria. Then we can continue to the next batch. Algorithm 2 gives a formal description.

---
**Algorithm 2** Multi-batched Algorithm With Delayed Feedback
---
1: **Require:** A multi-batched algorithm for the undelayed environment $\mathcal{ALG}$
2: Initialize dataset $\mathcal{D}^0 = \emptyset$
3: **for** batch $m = 1, ..., M$ **do**
4:     $\mathcal{ALG}$ calculates a policy $\pi^m$ (or a policy sequence) and a stopping criteria $SC$
5:     $\mathcal{D} = \emptyset$
6:     **while** $SC(\mathcal{D}) = 0$ **do**
7:         $k \leftarrow k + 1$
8:         In episode $k$, execute $\pi^m$
9:         Collect trajectory feedback in this batch that is observed by the end of the episode: $\{o^t : t + \tau_t = k\}$
10:         $\mathcal{D} = \mathcal{D} \cup \{o^t : t + \tau_t = k\}$
11:     **end while**
12:     Update dataset $\mathcal{D}^m = \mathcal{D}^{m-1} \cup \mathcal{D}$
13: **end for**
14: **Output:** A policy sequence $\{\pi^m\}$
---

For the performance of Algorithm 2, we have the following theorem, which relates the regret of delayed MSDM problems with the regret of multi-batched algorithms in the undelayed environment.

**Theorem 1.** *Assume that in the undelayed environment, we have a multi-batched algorithm with $N_b$ batches, and the regret of the algorithm in $K$ episodes can be upper bounded by $\widetilde{\mathrm{Regret}}(K)$ with probability at least $1 - \delta$. Then in the delayed feedback case, with probability at least $1 - \delta$, the regret can be upper bounded by*

$$\mathrm{Regret}(K) \leq \frac{1}{q}\widetilde{\mathrm{Regret}}(K) + \frac{2HN_b \log(K/\delta)}{q} + HN_b d_\tau(q) \tag{6}$$

*for any $q \in (0,1)$. In addition, if the delays satisfy Assumption 2, then with probability at least $1 - \delta$ the regret can be upper bounded by*

$$\text{Regret}(K) \leq \widetilde{\text{Regret}}(K) + HN_b\left(\mathbb{E}[\tau] + C_\tau\right), \tag{7}$$

*where $C_\tau = \min\{\sqrt{2v^2 \log(3KH/(2\delta))}, 2b \log(3KH/(2\delta))\}$ is a problem-independent constant.*

The proof is in Appendix A. The main idea is simple: in each batch, we can control the number of extra steps when waiting for the original algorithm to collect enough delayed feedback. And the extra regret can be bounded by the product of the number of batches and the number of extra steps per batch. Note that in Eq. (6) we do not need Assumption 2. This makes (6) more general.

Applying Theorem 1 to different settings, we will get various types of results. For example, in multi-armed bandit setting, using the Batched Successive Elimination (BaSE) algorithm in Gao et al. (2019), we can get a $O(\frac{1}{q}A \log(K) + d_\tau(q) \log(K))$ upper bound from (6), which recovers the SOTA result in Lancewicki et al. (2021) up to a $\log K$ factor in the delay-dependent term. We leave the details to Appendix C, and list the main results in Table 1.

# 5 Results for Markov Games

As an application, we study Markov games with delayed feedback using our framework. We first develop new multi-batched algorithms for Markov games, then combine them with our framework to obtain regret bounds for the delayed setting.

## 5.1 Tabular Zero-Sum Markov Game

First we study the tabular zero-sum Markov game setting. The algorithm is formally described in Algorithm 4 in Appendix D.

Our algorithm can be decomposed into two parts: an estimation step and a policy update step. The estimation step largely follows the method in Liu et al. (2021). For each $(h, s, a, b)$, we maintain an optimistic estimation $\bar{Q}_h^k(s, a, b)$ and a pessimistic estimation $\underline{Q}_h^k(s, a, b)$ of the underlying $Q_h^\star(s, a, b)$, by value iteration with bonus using the empirical estimate of the transition $\hat{\mathbb{P}}$. And we compute the Coarse Correlated Equilibrium (CCE) policy $\pi$ of the estimated value functions Xie et al. (2020). A CCE always exists, and can be computed efficiently.

In the policy update step, we choose the CCE policy w.r.t. the current estimated value function. Note that we use the doubling trick to control the number of updates: we only update the policy when there exists $(h, s, a, b)$ such that the visiting count $N_h(s, a, b)$ is doubled. This update scheme ensures that the batch number is small.

For the regret bound of Algorithm 4, we have the following theorem.

**Theorem 2.** *For any $p \in (0, 1]$, letting $\iota = \log(SABT/p)$, then with probability at least $1 - p$, the regret of Algorithm 4 satisfies*

$$\text{Regret}(K) \leq O\left(\sqrt{H^3 SABK\iota} + H^3 S^2 AB\iota^2\right).$$

*and the batch number is at most $O(HSAB \log K)$.*

The proof of Theorem 4 is similar to Theorem 4 in Liu et al. (2021), but the main difference is we have to deal with the sum of the Bernstein-type bonus under the doubling framework. To deal with this we borrow ideas from Zhang et al. (2021b). The whole proof is in Appendix D.

Adapting the algorithm to the delayed feedback setting using our framework, we have the following corollary.

**Corollary 1.** *For tabular Markov game with delayed feedback, running Algorithm 2 with $\mathcal{ALG}$ being Algorithm 4, we can upper bound the regret by*

$$\text{Regret}(K) \leq O\left(\frac{1}{q}\left(\sqrt{H^3 SABK\iota} + H^3 S^2 AB\iota^2\right) + H^2 SABd_\tau(q) \log K\right)$$

*for any $q \in (0, 1)$. In addition, if the delays satisfy Assumption 2, the regret can be upper bounded by*

$$\text{Regret}(K) \leq O\left(\sqrt{H^3 SABK\iota} + H^3 S^2 AB\iota^2 + H^2 SAB\mathbb{E}[\tau] \log K\right).$$

## 5.2 Linear Zero-Sum Markov Game

We consider the linear zero-sum Markov game setting, where the reward and transition have a linear structure.

**Assumption 3.** *For each $(x, a, b) \in \mathcal{S} \times \mathcal{A} \times \mathcal{B}$ and $h \in [H]$, we have*

$$r_h(x, a, b) = \phi(x, a, b)^\top \theta_h, \quad \mathbb{P}_h(\cdot \mid x, a, b) = \phi(x, a, b)^\top \mu_h(\cdot),$$

*where $\phi : \mathcal{S} \times \mathcal{A} \times \mathcal{A} \to \mathbb{R}^d$ is a known feature map, $\theta_h \in \mathbb{R}^d$ is an unknown vector and $\mu_h = (\mu_h^{(i)})_{i \in [d]}$ is a vector of $d$ unknown (signed) measures on $\mathcal{S}$. We assume that $\|\phi(\cdot, \cdot, \cdot)\| \leq 1, \|\theta_h\| \leq \sqrt{d}$ and $\|\mu_h(\mathcal{S})\| \leq \sqrt{d}$ for all $h \in [H]$, where $\| \cdot \|$ is the vector $\ell_2$ norm.*

The formal algorithm is in Algorithm 5 in Appendix D.2. It is also divided into two parts. The estimation step is similar to that in Xie et al. (2020), where we calculate an optimistic estimation $\bar{Q}_h^k(s, a, b)$ and a pessimistic estimation $\underline{Q}_h^k(s, a, b)$ of the underlying $Q_h^\star(s, a, b)$ by least square value iteration. And we compute the CCE policy $\pi$ of the estimated value functions.

In the policy update step, we update the policy to the CCE policy w.r.t. the current estimated value function. Note that instead of updating the policy in every episode, we only update it when there exists $h \in [H]$ such that $\det(\Lambda_h^k) > \eta \cdot \det(\Lambda_h)$. This condition implies that we have collected enough information at one direction. This technique is similar to Gao et al. (2021); Wang et al. (2021).

For the regret bound of Algorithm 5, we have the following theorem.

**Theorem 3.** *In Algorithm 5, let $\eta = 2$. Then we have*

$$\text{Regret}(K) \leq \tilde{O}(\sqrt{d^3 H^4 K}).$$

*And the batch number of Algorithm 5 is $\frac{dH}{\log 2} \log\left(1 + \frac{K}{\lambda}\right)$.*

*Proof.* See Appendix D.2 for a detailed proof. □

Adapting the algorithm to the delayed feedback setting using our framework, we have the following corollary.

**Corollary 2.** *For linear Markov game with delayed feedback, running Algorithm 2 with $\mathcal{ALG}$ being Algorithm 5, we can upper bound the regret by*

$$\text{Regret}(K) \leq \tilde{O}\left(\frac{1}{q}\sqrt{d^3 H^4 K} + dH^2 d_\tau(q) \log K\right)$$

*for any $q \in (0, 1)$. In addition, if the delays satisfy Assumption 2, the regret can be upper bounded by*

$$\text{Regret}(K) \leq \tilde{O}\left(\sqrt{d^3 H^4 K} + dH^2 \mathbb{E}[\tau] \log K\right).$$

## 5.3 Tabular General-Sum Markov Game

In this section, we introduce the multi-batched version of V-learning (Jin et al., 2021; Mao and Başar, 2022; Song et al., 2021) for general-sum Markov games. Our goal is to minimize the regret (2). The algorithm is formally described in Algorithm 6 in Appendix D.

The multi-batched V-learning algorithm maintains a value estimator $V_h(s)$, a counter $N_h(s)$, and a policy $\pi_h(\cdot \mid s)$ for each $s$ and $h$. We also maintain $S \times H$ different adversarial bandit algorithms. At step $h$ in episode $k$, the algorithm is divided into three steps: policy execution, $V$-value update, and policy update. In policy execution step, the algorithm takes action $a_h$ according to $\pi_h^{\tilde{k}}$, and observes reward $r_h$ and the next state $s_{h+1}$, and updates the counter $N_h(s_h)$. Note that $\pi_h^{\tilde{k}}$ is updated only when the visiting count of some state $N_h(s)$ doubles. Therefore it ensures a low batch number.

In the $V$-value update step, we update the estimated value function by

$$\tilde{V}_h(s_h) = (1 - \alpha_t)\tilde{V}_h(s_h) + \alpha_t\left(r_h + V_{h+1}(s_{h+1}) + \beta(t)\right),$$

where the learning rate is defined as

$$\alpha_t = \frac{H+1}{H+t}, \quad \alpha_t^0 = \prod_{j=1}^{t} \left(1 - \alpha_j\right), \quad \alpha_t^i = \alpha_i \prod_{j=i+1}^{t} \left(1 - \alpha_j\right).$$

and $\beta(t)$ is the bonus to promote optimism.

In the policy update step, the algorithm feeds the action $a_h$ and its "loss" $\frac{H - r_h + V_{h+1}(s_{h+1})}{H}$ to the $(s_h, h)^{\text{th}}$ adversarial bandit algorithm. Then it receives the updated policy $\pi_h(\cdot \mid s_h)$.

For two-player general sum Markov games, we let the two players run Algorithm 6 independently. Each player uses her own set of bonus that depends on the number of her actions. If we choose the adversarial bandit algorithm as Follow The Regularized Leader (FTRL) algorithm (Lattimore and Szepesvári, 2020), we have the following theorem for the regret of multi-batched V-learning.

**Theorem 4.** *Suppose we choose the adversarial bandit algorithm as FTRL. For any $\delta \in (0, 1)$ and $K \in \mathbb{N}$, let $\iota = \log(HSAK/\delta)$. Choose learning rate $\alpha_t$ and bonus $\{\beta(t)\}_{t=1}^{K}$ as $\beta(t) = c \cdot \sqrt{H^3 A\iota/t}$ so that $\sum_{i=1}^{t} \alpha_t^i \beta(i) = \Theta(\sqrt{H^3 A\iota/t})$ for any $t \in [K]$, where $A = \max_i A_i$ Then, with probability at least $1 - \delta$, after running Algorithm 6 for $K$ episodes, we have*

$$\text{Regret}(K) \leq O\left(\sqrt{H^5 SA\iota}\right).$$

*And the batch number is $O(nHS \log K)$, where $n$ is the number of players.*

*Proof.* See Appendix D for details. $\qquad\qquad\square$

Finally, adapting the algorithm to the delayed feedback setting using our framework, we have the following corollary.

**Corollary 3.** *For tabular general-sum Markov game with delayed feedback, running Algorithm 2 with $\mathcal{ALG}$ being Algorithm 6, we can upper bound the regret by*

$$\text{Regret}(K) \leq O\left(\frac{1}{q}\sqrt{H^5 SA\iota} + H^2 nSd_\tau(q) \log K\right)$$

*for any $q \in (0, 1)$. In addition, if the delays satisfy Assumption 2, the regret can be upper bounded by*

$$\text{Regret}(K) \leq O\left(\sqrt{H^5 SA\iota} + H^2 nS\mathbb{E}[\tau] \log K\right).$$

*Proof.* This corollary is directly implied by Theorems 1 and 4. $\qquad\qquad\square$

## 6   Conclusion and Future Work

In this paper we study delayed feedback in general multi-agent sequential decision making. We propose a novel reduction-based framework, which turns any multi-batched algorithm for sequential decision making with instantaneous feedback into a sample-efficient algorithm that can handle stochastic delays in sequential decision making. Based on this framework, we obtain a complete set of results for (multi-agent) sequential decision making with delayed feedback.

Our work has also shed light on future works. First, it remains unclear whether the result is tight for MDP and Markov game with delayed feedback. Second, it is possible that we can derive a multi-batched algorithm for tabular Markov game with a smaller batch number, since in tabular MDP, Zhang et al. (2022b) gives an algorithm with $O(H + \log_2 \log_2 K)$ batch number which is independent of $S$. It is not clear whether we can achieve similar results in MDP for Markov games. We believe a tighter result is possible in this setting with a carefully designed multi-batched algorithm.

## Acknowledgements

Liwei Wang is supported in part by NSF IIS 2110170, NSF DMS 2134106, NSF CCF 2212261, NSF IIS 2143493, NSF CCF 2019844, NSF IIS 2229881.

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

# A  Proof of Theorem 1

We first prove (7). First, since the delays are i.i.d. subexponential random variables, by standard concentration inequality and uniform bound Wainwright (2019), we have the following lemma:

**Lemma 5.** *Let $\{\tau_t\}_{t=1}^{\infty}$ be i.i.d $(v, b)$ subexponential random variables with expectation $\mathbb{E}[\tau]$. Then we have*

$$\mathbb{P}\left(\exists t \geq 1 : \tau_t \geq \mathbb{E}[\tau] + C_\tau\right) \leq 1 - \delta, \tag{8}$$

*where $C_\tau = \min\left\{\sqrt{2v^2 \log\left(\frac{3t}{2\delta}\right)}, 2b \log\left(\frac{3t}{2\delta}\right)\right\}$*

Define the good event $G = \{\exists t \geq 1 : \tau_t \geq \mathbb{E}[\tau] + C_\tau\}$. By Lemma 5, the good event happens with probability at least $1 - \delta$. Therefore, conditioned on the good event happens, in each batch of the algorithm, we only need to run $\mathbb{E}[\tau] + C_\tau$ episodes and the algorithm can observe enough feedback to finish this batch.

The regret in a batch can be divided into two parts: the first part is the regret caused by running the algorithm in the undelayed environment; the second part is caused by running extra episodes to wait for delayed feedback. The first part can be simply bounded by $\widetilde{\mathrm{Regret}}(K)$. For the second part, since each episode contributes at most $H$ regret, the total regret can be bounded by

$$\mathrm{Regret}(K) \leq \widetilde{\mathrm{Regret}}(K) + HN_b\left(\mathbb{E}[\tau] + C_\tau\right). \tag{9}$$

For the second inequality, we have the following lemma which reveals the relation between the number of observed feedback and the actual number of episodes using quantile function.

**Lemma 6.** *Denote the number of observed feedback at time $t$ as $n_t$. For any quantile $q$ and any $t$, we have*

$$\Pr\left[n_{t+d_\tau(q)} < \frac{q}{2}t\right] \leq \exp\left(-\frac{q}{8}t\right). \tag{10}$$

*Proof.* Define $\mathbb{I}\left\{\tau_s \leq d_\tau(q)\right\}$ to be an indicator that on time $s$ that the delay is smaller than $d_\tau(q)$. Note that $\mathbb{E}\left[\mathbb{I}\left\{\tau_s \leq d_\tau(q)\right\}\right] \geq q$. Thus,

$$\Pr\left[n_{t+d_\tau(q)} < \frac{q}{2}t\right] \leq \Pr\left[\sum_{s=1}^{t} \mathbb{I}\left\{\tau_s \leq d_\tau(q)\right\} < \frac{q}{2}t\right]$$

$$\leq \Pr\left[\sum_{s=1}^{t} \mathbb{I}\left\{\tau_s \leq d_\tau(q)\right\} < \frac{1}{2}\sum_{s=1}^{t} \mathbb{E}\left[\mathbb{I}\left\{\tau_s \leq d_\tau(q)\right\}\right]\right] \tag{11}$$

$$\leq \exp\left(-\frac{1}{8}\sum_{s=1}^{t} \mathbb{E}\left[\mathbb{I}\left\{\tau_s \leq d_\tau(q)\right\}\right]\right) \leq \exp\left(-\frac{q}{8}t\right),$$

where the third inequality follows from the relative Chernoff bound, and the last inequality is since $\sum_{s=1}^{t} \mathbb{E}\left[\mathbb{I}\left\{\tau_s \leq d_\tau(q)\right\}\right] \geq q \cdot t$. □

Lemma 6 means, if the length of the batch in the undelayed environment $m > \frac{\log(K/\delta)}{q}$, then with probability at least $1 - \frac{\delta}{K}$, if we run the batch for $\frac{m}{q} + d_\tau(q)$ episodes, we will observe at least $m$ feedbacks, which is enough for the algorithm to finish this batch. On the other hand, if $m \leq \frac{\log(K/\delta)}{q}$, then the regret of this batch can be trivially bounded by $\frac{H \log(K/\delta)}{q}$. Hence the total regret can be bounded by

$$\mathrm{Regret}(K) \leq \mathrm{Regret}\left(\frac{K}{q} + d_\tau(q)N_b\right) \leq \frac{1}{q}\widetilde{\mathrm{Regret}}(K) + \frac{2HN_b \log(K/\delta)}{q} + HN_b d_\tau(q), \tag{12}$$

which finishes the proof.

# B  Details of Algorithms for Linear Bandits

The phase elimination algorithm for linear bandits in Lattimore and Szepesvári (2020) is as follows. This algorithm is naturally a multi-batched algorithm, with each phase as one batch. Since the length of phase $\ell$ is $O(2^\ell)$, the number of phases is $O(\log K)$. Note that in each phase the policy is pre-calculated at the beginning of the phase, and does not rely on the data of current phase.

---

**Algorithm 3** Phase Elimination

---

1: **Input:** $\mathcal{A} \subset \mathbb{R}^d$, $\beta > 1$, $\delta$
2: Set $\ell = 1$ and let $\mathcal{A}_1 = \mathcal{A}$
3: Let $t_\ell = t$ be the current timestep and find $G$-optimal design $\pi_\ell \in \mathcal{P}(\mathcal{A}_\ell)$ with $\mathrm{Supp}(\pi_\ell) \leq d(d+1)/2$ that maximises

$$\log \det V(\pi_\ell) \text{ subject to } \sum_{a \in \mathcal{A}_\ell} \pi_\ell(a) = 1$$

4: Let $\varepsilon_\ell = 2^{-\ell}$ and

$$T_\ell(a) = \left\lceil \frac{2d\pi_\ell(a)}{\varepsilon_\ell^2} \log\left(\frac{k\ell(\ell+1)}{\delta}\right) \right\rceil \text{ and } T_\ell = \sum_{a \in \mathcal{A}_\ell} T_\ell(a)$$

5: Choose each action $a \in \mathcal{A}_\ell$ exactly $T_\ell(a)$ times.
6: Calculate the empirical estimate:

$$\hat{\theta}_\ell = V_\ell^{-1} \sum_{t=t_\ell}^{t_\ell+T_\ell} A_t X_t \quad \text{with} \quad V_\ell = \sum_{a \in \mathcal{A}_\ell} T_\ell(a) aa^\top$$

using observable data in this phase.
7: Eliminating low reward arms:

$$\mathcal{A}_{\ell+1} = \left\{ a \in \mathcal{A}_\ell : \max_{b \in \mathcal{A}_\ell} \left\langle \hat{\theta}_\ell, b - a \right\rangle \leq 2\varepsilon_\ell \right\}.$$

8: $\ell \leftarrow \ell + 1$ and go to Line 2.

---

**Theorem 7** (Theorem 22.1 in Lattimore and Szepesvári (2020)). *With probability at least $1 - \delta$, the regret of Algorithm 5 satisfies*

$$\mathrm{Regret}(K) \leq C\sqrt{Kd \log\left(\frac{A \log(K)}{\delta}\right)}, \tag{13}$$

*where $C > 0$ is a universal constant. If $\delta = O(1/K)$, then $\mathbb{E}\left[\mathrm{Regret}(K)\right] \leq C\sqrt{Kd \log(AK)}$ for an appropriately chosen universal constant $C > 0$.*

For infinite action set $\mathcal{A} = [-1, 1]^d$, we use a covering technique: we find a $\epsilon$-covering of the action space, and run Phase Elimination on the covering set. The covering number of the set $[-1, 1]^d$ is $O(d \log(1/\epsilon))$. And the error caused by using $\epsilon$-covering will lead to a $O(\epsilon K)$ increase in regret. Therefore, by setting $\epsilon = \frac{1}{K}$, the regret of the whole algorithm is $O(d\sqrt{K} \log K)$.

## C  Results for Bandits and MDPs

We now apply Theorem 1 to different bandit and MDP problems.

In multi-armed bandit settings, using the BaSE algorithm Gao et al. (2019), we can get the following bound.

**Corollary 4.** *For multi-armed bandit with delayed feedback, running Algorithm 2 with $\mathcal{ALG}$ being the BaSE algorithm in Gao et al. (2019), we can upper bound the regret by*

$$\text{Regret}(K) \leq O\left(\frac{A \log K}{q \min_{i \neq \star} \Delta_i} + \frac{\log^2 K}{q} + d_\tau(q) \log K\right)$$

*for any $q \in (0, 1)$, where $\Delta_i$ is the optimal gap of the $i^{th}$ arm. In addition, if the delays satisfy Assumption 2, the regret can be upper bounded by*

$$\text{Regret}(K) \leq O\left(\frac{A \log K}{\min_{i \neq \star} \Delta_i} + \mathbb{E}[\tau] \log K\right).$$

This result recovers the SOTA result in Lancewicki et al. (2021) up to a $\log K$ factor in the delay-dependent term.

In linear bandit settings, using the Phase Elimination algorithm (Lattimore and Szepesvári, 2020), we can get the following bound.

**Corollary 5.** *For linear bandit with delayed feedback, let $d$ be the dimension of the feature space. When the arm set $\mathcal{A}$ is finite, running Algorithm 2 with $\mathcal{ALG}$ being the Phase Elimination algorithm in Lattimore and Szepesvári (2020), we can upper bound the regret by*

$$\text{Regret}(K) \leq O\left(\frac{1}{q}\sqrt{dK \log A} + \frac{\log^2 K}{q} + d_\tau(q) \log K\right)$$

*for any $q \in (0, 1)$. In addition, if the delays satisfy Assumption 2, the regret can be upper bounded by*

$$\text{Regret}(K) \leq O\left(\sqrt{dK \log A} + \mathbb{E}[\tau] \log K\right).$$

When the action set is infinitely large (for example, $\mathcal{A} = [-1, 1]^d$), we can use techniques such as covering to obtain regret bounds whose main term scales as $\tilde{O}(d\sqrt{K})$. Compared with the result in Howson et al. (2021) which scales as $\tilde{O}(d\sqrt{K} + d^{3/2}\mathbb{E}[\tau])$, our bound is tighter in the delay-dependent term as we remove the $d^{3/2}$ factor.

In tabular MDP settings, using the policy elimination algorithm Zhang et al. (2022b), we can get the following bound.

**Corollary 6.** *For tabular MDP with delayed feedback, running Algorithm 2 with $\mathcal{ALG}$ being the policy elimination algorithm in Zhang et al. (2022b), we can upper bound the regret by*

$$\text{Regret}(K) \leq O\left(\frac{1}{q}\sqrt{SAH^3K\iota} + \frac{(H + \log\log K)\log K}{q} + (H + \log\log K)d_\tau(q)\right)$$

*for any $q \in (0, 1)$, where $\iota$ is some log factor. In addition, if the delays satisfy Assumption 2, the regret can be upper bounded by*

$$\text{Regret}(K) \leq O\left(\sqrt{SAH^3K\iota} + H(H + \log\log K)\mathbb{E}[\tau]\right).$$

Compared with the previous result $\sqrt{H^3SAK\iota} + H^2SA\mathbb{E}[\tau]\log K$ in Howson et al. (2022), our bound removes the $SA$ factor in the delay-dependent term, therefore is tighter.

We can also apply our framework to linear MDP and MDP with general function approximation settings with delay. These settings have not been studied before, therefore our result is novel. In linear MDP settings, using the algorithm in Gao et al. (2021) we can get the following bound.

**Corollary 7.** *For linear MDP with delayed feedback, let $d$ be the dimension of the feature space. Running Algorithm 2 with $\mathcal{ALG}$ being the algorithm in Gao et al. (2021), we can upper bound the regret by*

$$\text{Regret}(K) \leq \tilde{O}\left(\frac{1}{q}\sqrt{d^3H^4K} + dH^2d_\tau(q)\log K\right)$$

*for any $q \in (0, 1)$. In addition, if the delays satisfy Assumption 2, the regret can be upper bounded by*

$$\text{Regret}(K) \leq \tilde{O}\left(\sqrt{d^3H^4K} + dH^2\mathbb{E}[\tau]\log K\right).$$

And in MDP with general function approximation, using the algorithm in Kong et al. (2021) we get the following bound.

**Corollary 8.** *For MDP with general function approximation and delayed feedback, let* $\dim_E(\mathcal{F}, \epsilon)$ *be the $\epsilon$-eluder dimension of the function class $\mathcal{F}$. Running Algorithm 2 with $\mathcal{ALG}$ being the algorithm in Kong et al. (2021), we can upper bound the regret by*

$$\mathrm{Regret}(K) \leq \tilde{O}\left(\frac{1}{q}\sqrt{\dim_E^2(\mathcal{F}, 1/K)H^4 K} + H^2 \dim_E(\mathcal{F}, 1/K)d_\tau(q)\right)$$

*for any $q \in (0, 1)$. In addition, if the delays satisfy Assumption 2, the regret can be upper bounded by*

$$\mathrm{Regret}(K) \leq \tilde{O}\left(\sqrt{\dim_E^2(\mathcal{F}, 1/K)H^4 K} + H^2 \dim_E(\mathcal{F}, 1/K)\mathbb{E}[\tau]\right).$$

# D  Proofs for Tabular Markov Games

## D.1  Multi-batched Optimistic Nash-VI

---

**Algorithm 4** Multi-batched Algorithm for Tabular Markov Game

---

1: **Initialize:** for any $(s, a, b, h), \bar{Q}_h(s, a, b) \leftarrow H, \underline{Q}_h(s, a, b) \leftarrow 0, \Delta \leftarrow H, N_h(s, a, b) \leftarrow 0,$ trigger set $\mathcal{L} \leftarrow \left\{2^{i-1} \mid 2^i \leq KH, i = 1, 2, \ldots\right\}$
2: **for** episode $k = 1, 2, \cdots, K$ **do**
3:     **if** there exists $(h, s, a, b)$ such that $N_h^k(s, a, b) \in \mathcal{L}$ **then**
4:         **for** step $h = H, H-1, ..., 1$ **do**
5:             **for** $(s, a, b) \in \mathcal{S} \times \mathcal{A} \times \mathcal{B}$ **do**
6:                 $t \leftarrow N_h^k(s, a, b)$
7:                 **if** $t > 0$ **then**
8:                     $\beta_h^k \leftarrow \mathrm{BONUS}\left(t, \widehat{\mathbb{V}}_h\left[\left(\bar{V}_{h+1} + \underline{V}_{h+1}\right)/2\right](s, a, b)\right)$
9:                     $\gamma_h^k \leftarrow (c/H)\widehat{\mathbb{P}}_h\left(\bar{V}_{h+1} - \underline{V}_{h+1}\right)(s, a, b)$
10:                     $\bar{Q}_h^k(s, a, b) \leftarrow \min\left\{\left(r_h + \widehat{\mathbb{P}}_h\bar{V}_{h+1}\right)(s, a, b) + \gamma_h^k + \beta_h^k, H\right\}$
11:                     $\underline{Q}_h^k(s, a, b) \leftarrow \max\left\{\left(r_h + \widehat{\mathbb{P}}_h\underline{V}_{h+1}\right)(s, a, b) - \gamma_h^k - \beta_h^k, 0\right\}$
12:                 **end if**
13:             **end for**
14:             **for** $s \in \mathcal{S}$ **do**
15:                 $\pi_h^k(\cdot, \cdot \mid s) \leftarrow \mathrm{CCE}\left(\bar{Q}_h(s, \cdot, \cdot), \underline{Q}_h(s, \cdot, \cdot)\right)$
16:                 $\bar{V}_h(s) \leftarrow \left(\mathbb{E}_{\pi_h}\bar{Q}_h\right)(s); \quad \underline{V}_h(s) \leftarrow \left(\mathbb{E}_{\pi_h}\underline{Q}_h\right)(s)$
17:             **end for**
18:         **end for**
19:     **end if**
20:     **for** step $h = 1, ..., H$ **do**
21:         take action $(a_h, b_h) \sim \pi_h^{\tilde{k}}(\cdot, \cdot \mid s_h)$, observe reward $r_h$ and next state $s_{h+1}$.
22:         add 1 to $N_h^k(s_h, a_h, b_h)$ and $N_h^k(s_h, a_h, b_h, s_{h+1})$.
23:         $\widehat{\mathbb{P}}_h(\cdot \mid s_h, a_h, b_h) \leftarrow N_h^k(s_h, a_h, b_h, \cdot)/N_h^k(s_h, a_h, b_h)$
24:     **end for**
25: **end for**

---

In this part, we list some of the important lemmas for the tabular Markov game algorithm, most of which are proven in the previous literature Liu et al. (2021). These lemmas are very useful for proving the regret bound in our main theorem.

We denote $V^k, Q^k, \pi^k, \mu^k$ and $\nu^k$ for values and policies at the beginning of the $k$-th episode. In particular, $N_h^k(s, a, b)$ is the number we have visited the state-action tuple $(s, a, b)$ at the $h$-th step

before the $k$-th episode. $N_h^k(s, a, b, s')$ is defined by the same token. Using this notation, we can further define the empirical transition by $\widehat{\mathbb{P}}_h^k(s' \mid s, a, b) := N_h^k(s, a, b, s') / N_h^k(s, a, b)$. If $N_h^k(s, a, b) = 0$, we set $\widehat{\mathbb{P}}_h^k(s' \mid s, a, b) = 1/S$.

We define the empirical variance operator

$$\widehat{\mathbb{V}}_h^k V(s, a, b) := \text{Var}_{s' \sim \widehat{\mathbb{P}}_h^k(\cdot \mid s, a, b)} V(s') \tag{14}$$

and true variance operator

$$\mathbb{V}_h^k V(s, a, b) := \text{Var}_{s' \sim \mathbb{P}_h^k(\cdot \mid s, a, b)} V(s'). \tag{15}$$

The bonus terms can be written as

$$b_h^k(s, a, b) = \beta_h^k(s, a, b) + \gamma_h^k(s, a, b) \tag{16}$$

where

$$\beta_h^k(s, a, b) := C \left( \sqrt{\frac{\widehat{\boldsymbol{V}}_h^k \left[ \left( \bar{V}_{h+1}^k + \underline{V}_{h+1}^k \right) / 2 \right](s, a, b)}{\max \left\{ N_h^k(s, a, b), 1 \right\}}} + \frac{H^2 S \iota}{\max \left\{ N_h^k(s, a, b), 1 \right\}} \right) \tag{17}$$

and the lower-order term

$$\gamma_h^k(s, a, b) := \frac{C}{H} \widehat{\mathbb{P}}_h \left( \bar{V}_{h+1}^k - \underline{V}_{h+1}^k \right)(s, a, b) \tag{18}$$

for some absolute constant $C > 0$.

The following lemma is the standard concentration result.

**Lemma 8** (Lemma 21, Liu et al. (2021)). *Let $c_1$ be some large absolute constant. Define event $E_1$ to be: for all $h, s, a, b, s'$ and $k \in [K]$,*

$$\begin{cases} \left| \left[ \left( \widehat{\mathbb{P}}_h^k - \mathbb{P}_h \right) V_{h+1}^\star \right](s, a, b) \right| \le c_1 \left( \sqrt{\frac{\widehat{\mathbb{V}}_h^k V_{h+1}^\star(s, a, b) \iota}{\max\{N_h^k(s, a, b), 1\}}} + \frac{H\iota}{\max\{N_h^k(s, a, b), 1\}} \right) \\ \left| \left( \widehat{\mathbb{P}}_h^k - \mathbb{P}_h \right)(s' \mid s, a, b) \right| \le c_1 \left( \sqrt{\frac{\min\left\{ \mathbb{P}_h(s' \mid s, a, b), \widehat{\mathbb{P}}_h^k(s' \mid s, a, b) \right\} \iota}{\max\{N_h^k(s, a, b), 1\}}} + \frac{\iota}{\max\{N_h^k(s, a, b), 1\}} \right) \\ \left\| \left( \widehat{\mathbb{P}}_h^k - \mathbb{P}_h \right)(\cdot \mid s, a, b) \right\|_1 \le c_1 \sqrt{\frac{S\iota}{\max\{N_h^k(s, a, b), 1\}}}. \end{cases} \tag{19}$$

*We have $\mathbb{P}(E_1) \ge 1 - p$.*

The following lemma shows the upper and lower bounds are indeed the upper and lower bounds of the best responses.

**Lemma 9** (Lemma 22, Liu et al. (2021)). *Suppose event $E_1$ holds. Then for all $h, s, a, b$ and $k \in [K]$, we have*

$$\begin{cases} \bar{Q}_h^k(s, a, b) \ge Q_h^{\dagger, \nu^k}(s, a, b) \ge Q_h^{\mu^k, \dagger}(s, a, b) \ge \underline{Q}_h^k(s, a, b), \\ \bar{V}_h^k(s) \ge V_h^{\dagger, \nu^k}(s) \ge V_h^{\mu^k, \dagger}(s) \ge \underline{V}_h^k(s). \end{cases} \tag{20}$$

**Lemma 10** (Lemma 23, Liu et al. (2021)). *Suppose event $E_1$ holds. Then for all $h, s, a, b$ and $k \in [K]$, we have*

$$\left| \widehat{\mathbb{V}}_h^k \left[ \left( \bar{V}_{h+1}^k + \underline{V}_{h+1}^k \right) / 2 \right] - \mathbb{V}_h V_{h+1}^{\pi^k} \right| (s, a, b)$$

$$\le 4H \mathbb{P}_h \left( \bar{V}_{h+1}^k - \underline{V}_{h+1}^k \right)(s, a, b) + O \left( 1 + \frac{H^4 S \iota}{\max \left\{ N_h^k(s, a, b), 1 \right\}} \right). \tag{21}$$

*Proof of Theorem 4.* Suppose event $E_1$ happens. We define

$$
\begin{cases}
\Delta_h^k := \left( \bar{V}_h^k - \underline{V}_h^k \right) \left( s_h^k \right) \\
\zeta_h^k := \Delta_h^k - \left( \bar{Q}_h^k - \underline{Q}_h^k \right) \left( s_h^k, a_h^k, b_h^k \right) \\
\xi_h^k := \mathbb{P}_h \left( \bar{V}_{h+1}^k - \underline{V}_{h+1}^k \right) \left( s_h^k, a_h^k, b_h^k \right) - \Delta_{h+1}^k .
\end{cases}
\tag{22}
$$

Using standard decomposition techniques we have

$$
\begin{aligned}
\Delta_h^k &= \zeta_h^k + \left( \bar{Q}_h^k - \underline{Q}_h^k \right) \left( s_h^k, a_h^k, b_h^k \right) \\
&\leq \zeta_h^k + 2\beta_h^k + 2\gamma_h^k + \widehat{\mathbb{P}}_h^k \left( \bar{V}_{h+1}^k - \underline{V}_{h+1}^k \right) \left( s_h^k, a_h^k, b_h^k \right) \\
&\leq \zeta_h^k + 2\beta_h^k + 2\gamma_h^k + \mathbb{P}_h \left( \bar{V}_{h+1}^k - \underline{V}_{h+1}^k \right) \left( s_h^k, a_h^k, b_h^k \right) + c_2 \left( \frac{\mathbb{P}_h \left( \bar{V}_{h+1}^k - \underline{V}_{h+1}^k \right) \left( s_h^k, a_h^k, b_h^k \right)}{H} + \frac{H^2 S \iota}{\max\left\{ N_h^k, 1 \right\}} \right) \\
&\leq \zeta_h^k + 2\beta_h^k + \mathbb{P}_h \left( \bar{V}_{h+1}^k - \underline{V}_{h+1}^k \right) \left( s_h^k, a_h^k, b_h^k \right) + 2c_2 C \left( \frac{\mathbb{P}_h \left( \bar{V}_{h+1}^k - \underline{V}_{h+1}^k \right) \left( s_h^k, a_h^k, b_h^k \right)}{H} + \frac{H^2 S \iota}{\max\left\{ N_h^k, 1 \right\}} \right) \\
&\leq \zeta_h^k + \left( 1 + \frac{c_3}{H} \right) \mathbb{P}_h \left( \bar{V}_{h+1}^k - \underline{V}_{h+1}^k \right) \left( s_h^k, a_h^k, b_h^k \right) \\
&\quad + 4c_2 C \left( \sqrt{ \frac{\iota \widehat{\mathbb{V}}_h^k \left[ \left( \bar{V}_{h+1}^k + \underline{V}_{h+1}^k \right) / 2 \right] \left( s_h^k, a_h^k, b_h^k \right)}{\max\left\{ N_h^k \left( s_h^k, a_h^k, b_h^k \right), 1 \right\}} } + \frac{H^2 S \iota}{\max\left\{ N_h^k \left( s_h^k, a_h^k, b_h^k \right), 1 \right\}} \right) .
\end{aligned}
\tag{23}
$$

By Lemma 10,

$$
\begin{aligned}
&\sqrt{ \frac{\iota \widehat{\mathbb{V}}_h^k \left[ \left( \bar{V}_{h+1}^k + \underline{V}_{h+1}^k \right) / 2 \right] (s, a, b)}{\max\left\{ N_h^k(s, a, b), 1 \right\}} } \\
&\leq \left( \sqrt{ \frac{\iota \mathbb{V}_h V_{h+1}^{\pi^k}(s, a, b) + \iota}{\max\left\{ N_h^k(s, a, b), 1 \right\}} } + \sqrt{ \frac{H \iota \mathbb{P}_h \left( \bar{V}_{h+1}^k - \underline{V}_{h+1}^k \right) (s, a, b)}{\max\left\{ N_h^k(s, a, b), 1 \right\}} } + \frac{H^2 \sqrt{S} \iota}{\max\left\{ N_h^k(s, a, b), 1 \right\}} \right) \\
&\leq c_4 \left( \sqrt{ \frac{\iota \mathbb{V}_h V_{h+1}^{\pi^k}(s, a, b) + \iota}{\max\left\{ N_h^k(s, a, b), 1 \right\}} } + \frac{\mathbb{P}_h \left( \bar{V}_{h+1}^k - \underline{V}_{h+1}^k \right) (s, a, b)}{H} + \frac{H^2 \sqrt{S} \iota}{\max\left\{ N_h^k(s, a, b), 1 \right\}} \right) ,
\end{aligned}
\tag{24}
$$

where $c_4$ is some absolute constant. Define $c_5 := 4c_2 c_4 C + c_3$ and $\kappa := 1 + c_5 / H$. Now we have

$$
\Delta_h^k \leq \kappa \Delta_{h+1}^k + \kappa \xi_h^k + \zeta_h^k + O \left( \sqrt{ \frac{\iota \mathbb{V}_h V_{h+1}^{\pi^k} \left( s_h^k, a_h^k, b_h^k \right)}{N_h^k \left( s_h^k, a_h^k, b_h^k \right)} } + \sqrt{ \frac{\iota}{N_h^k \left( s_h^k, a_h^k, b_h^k \right)} } + \frac{H^2 S \iota}{N_h^k \left( s_h^k, a_h^k, b_h^k \right)} \right) \Bigg\} .
\tag{25}
$$

Recursing this argument for $h \in [H]$ and summing over $k$,

$$
\sum_{k=1}^{K} \Delta_1^k \leq \sum_{k=1}^{K} \sum_{h=1}^{H} \left[ \kappa^{h-1} \zeta_h^k + \kappa^h \xi_h^k + O \left( \sqrt{ \frac{\iota \mathbb{V}_h V_{h+1}^{\pi^k} \left( s_h^k, a_h^k, b_h^k \right)}{\max\left\{ N_h^k, 1 \right\}} } + \sqrt{ \frac{\iota}{\max\left\{ N_h^k, 1 \right\}} } + \frac{H^2 S \iota}{\max\left\{ N_h^k, 1 \right\}} \right) \right] .
\tag{26}
$$

It is easy to check that $\{\zeta_h^k\}$ and $\{\xi_h^k\}$ are martingales. By Azuma-Hoeffding inequality, with probability at least $1 - p$,

$$
\begin{cases}
\sum_{k=1}^{K} \sum_{h=1}^{H} \kappa^{h-1} \zeta_h^k \le O(H\sqrt{HK\iota}) \\
\sum_{k=1}^{K} \sum_{h=1}^{H} \kappa^{h} \xi_h^k \le O(H\sqrt{HK\iota}).
\end{cases}
\tag{27}
$$

Now we deal with the rest terms. Note that under the doubling epoch update framework, despite those episodes in which an update is triggered, the number of visits of $(s, a)$ between the $i$-th update of $\hat{P}_{s,a}$ and the $i+1$-th update of $\hat{P}_{s,a}$ do not exceed $2^{i-1}$.

We define $i_{\max} = \max\left\{i \mid 2^{i-1} \le KH\right\} = \lfloor \log_2(KH) \rfloor + 1$. To facilitate the analysis, we first derive a general deterministic result. Let $\mathcal{K}$ be the set of indexes of episodes in which no update is triggered. By the update rule we have $\left|\mathcal{K}^C\right| \le SA\left(\log_2(KH) + 1\right)$. Let $h_0(k)$ be the first time an update is triggered in the $k$-th episode if there is an update in this episode and otherwise $H + 1$. Define $\mathcal{X} = \left\{(k, h) \mid k \in \mathcal{K}^C, h_0(k) + 1 \le h \le H\right\}$.

Let $w = \left\{w_h^k \ge 0 \mid 1 \le h \le H, 1 \le k \le K\right\}$ be a group of non-negative weights such that $w_h^k \le 1$ for any $(k, h) \in [H] \times [K]$ and $w_h^k = 0$ for any $(k, h) \in \mathcal{X}$. As in Zhang et al. (2021b), We can calculate

$$
\sum_{k=1}^{K} \sum_{h=1}^{H} \sqrt{\frac{w_h^k}{N_h^k\left(s_h^k, a_h^k\right)}}
$$

$$
\le \sum_{k=1}^{K} \sum_{h=1}^{H} \sum_{s,a} \sum_{i=3}^{i_{\max}} \mathbb{I}\left[\left(s_h^k, a_h^k\right) = (s, a), N_h^k(s, a) = 2^{i-1}\right] \sqrt{\frac{w_h^k}{2^{i-1}}} + 8SA\left(\log_2(KH) + 4\right)
$$

$$
= \sum_{s,a} \sum_{i=3}^{i_{\max}} \frac{1}{\sqrt{2^{i-1}}} \sum_{k=1}^{K} \sum_{h=1}^{H} \mathbb{I}\left[\left(s_h^k, a_h^k\right) = (s, a), N_h^k(s, a) = 2^{i-1}\right] \sqrt{w_h^k} + 8SA\left(\log_2(KH) + 4\right)
$$

$$
\le \sum_{s,a} \sum_{i=3}^{i_{\max}} \sqrt{\frac{\sum_{k=1}^{K} \sum_{h=1}^{H} \mathbb{I}\left[\left(s_h^k, a_h^k\right) = (s, a), N_h^k(s, a) = 2^{i-1}\right]}{2^{i-1}}} \cdot
$$

$$
\sqrt{\left(\sum_{k=1}^{K} \sum_{h=1}^{H} \mathbb{I}\left[\left(s_h^k, a_h^k\right) = (s, a), N_h^k(s, a) = 2^{i-1}\right] w_h^k\right)} + 8SA\left(\log_2(KH) + 4\right)
$$

$$
\le \sqrt{SAi_{\max} \sum_{k=1}^{K} \sum_{h=1}^{H} w_h^k} + 8SA\left(\log_2(KH) + 4\right).
$$

(28)

By plugging $w_h^k = \mathbb{V}_h V_{h+1}^{\pi^k}\left(s_h^k, a_h^k, b_h^k\right) \mathbb{I}[(k, h) \notin \mathcal{X}]$ and $w_h^k = \mathbb{I}[(k, h) \notin \mathcal{X}]$ we have

$$
\sum_{k=1}^{K} \sum_{h=1}^{H} \left[O\left(\sqrt{\frac{\iota \mathbb{V}_h V_{h+1}^{\pi^k}\left(s_h^k, a_h^k, b_h^k\right)}{\max\left\{N_h^k, 1\right\}}} + \sqrt{\frac{\iota}{\max\left\{N_h^k, 1\right\}}} + \frac{H^2 S \iota}{\max\left\{N_h^k, 1\right\}}\right)\right]
$$

$$
\le O\left(\sqrt{SAi_{\max}\iota \sum_{k=1}^{K} \sum_{h=1}^{H} \mathbb{V}_h V_{h+1}^{\pi^k}\left(s_h^k, a_h^k, b_h^k\right) \mathbb{I}[(k, h) \notin \mathcal{X}]} + \sqrt{SAi_{\max} \sum_{k=1}^{K} \sum_{h=1}^{H} \mathbb{I}[(k, h) \notin \mathcal{X}]\iota + S^2 A \iota \log_2(KH)} + \left|\mathcal{K}\right.\right.
$$

$$
\le O\left(\sqrt{SAi_{\max}\iota \sum_{k=1}^{K} \sum_{h=1}^{H} \mathbb{V}_h V_{h+1}^{\pi^k}\left(s_h^k, a_h^k, b_h^k\right) \mathbb{I}[(k, h) \notin \mathcal{X}]} + \sqrt{SAi_{\max}K\iota} + S^2 A \iota \log_2(KH)\right) + \left|\mathcal{K}^C\right|
$$

(29)

By the Law of total variation and standard martingale concentration (see Lemma C.5 in Jin et al. (2018) for a formal proof), with probability at least $1 - p$, we have

$$\sum_{k=1}^{K} \sum_{h=1}^{H} \mathbb{V}_h V_{h+1}^{\pi^k} \left( s_h^k, a_h^k, b_h^k \right) \leq O \left( H^2 K + H^3 \iota \right). \tag{30}$$

By Pigeon-hole argument,

$$\sum_{k=1}^{K} \sum_{h=1}^{H} \frac{1}{\sqrt{\max \left\{ N_h^k, 1 \right\}}} \leq \sum_{s,a,b,h: N_h^K(s,a,b) > 0} \sum_{n=1}^{N_h^K(s,a,b)} \frac{1}{\sqrt{n}} + HSAB \leq O(\sqrt{H^2 SABK} + HSAB) \tag{31}$$

$$\sum_{k=1}^{K} \sum_{h=1}^{H} \frac{1}{\max \left\{ N_h^k, 1 \right\}} \leq \sum_{s,a,b,h: N_h^K(s,a,b) > 0} \sum_{n=1}^{N_h^K(s,a,b)} \frac{1}{n} + HSAB \leq O(HSAB\iota). \tag{32}$$

Putting all relations together, we obtain that

$$\text{Regret}(K) = \sum_{k=1}^{K} \left( V_1^{\dagger, \nu^k} - V_1^{\mu^k, \dagger} \right)(s_1) \leq O \left( \sqrt{H^3 SABK\iota} + H^3 S^2 AB\iota^2 \right) \tag{33}$$

which completes the proof. □

## D.2 Multi-batched Algorithms for Linear Markov Games

---

**Algorithm 5** Multi-batched Algorithm for Linear Markov Games

---

1: **Require:** regularization parameter $\lambda$
2: **Initialize:** $\Lambda_h = \Lambda_h^0 = \lambda I_d$
3: **for** episode $k = 1, 2, \cdots, K$ **do**
4:    $\Lambda_h^k = \sum_{\tau=1}^{k-1} \phi(s_h^\tau, a_h^\tau, b_h^\tau) \phi(s_h^\tau, a_h^\tau, b_h^\tau)^\top + \lambda I_d$
5: **end for**
6: **if** $\exists h \in [H], \det(\Lambda_h^k) > \eta \cdot \det(\Lambda_h)$ **then**
7:    $\bar{V}_{H+1}^k \leftarrow 0, \underline{V}_{H+1}^k \leftarrow 0$
8:    **for** step $h = H, H-1, \cdots, 1$ **do**
9:       $\Lambda_h \leftarrow \Lambda_h^k$
10:       $\bar{w}_h^k \leftarrow (\Lambda_h^k)^{-1} \sum_{\tau=1}^{k-1} \phi(s_h^\tau, a_h^\tau, b_h^\tau) \cdot [r_h(s_h^\tau, a_h^\tau, b_h^\tau) + \bar{V}_{h+1}^k(s_{h+1}^\tau)]$
11:       $\underline{w}_h^k \leftarrow (\Lambda_h^k)^{-1} \sum_{\tau=1}^{k-1} \phi(s_h^\tau, a_h^\tau, b_h^\tau) \cdot [r_h(s_h^\tau, a_h^\tau, b_h^\tau) + \underline{V}_{h+1}^k(s_{h+1}^\tau)]$
12:       $\Gamma_h^k(\cdot, \cdot, \cdot) \leftarrow \beta \sqrt{\phi(\cdot, \cdot, \cdot)^\top (\Lambda_h^k)^{-1} \phi(\cdot, \cdot, \cdot)}$
13:       $\bar{Q}_h^k(\cdot, \cdot, \cdot) \leftarrow \Pi_H \{ (\bar{w}_h^k)^\top \phi(\cdot, \cdot, \cdot) + \Gamma_h^k(\cdot, \cdot, \cdot) \}$
14:       $\underline{Q}_h^k(\cdot, \cdot, \cdot) \leftarrow \Pi_H \{ (\underline{w}_h^k)^\top \phi(\cdot, \cdot, \cdot) - \Gamma_h^k(\cdot, \cdot, \cdot) \}$
15:       For each $s$, let $\pi_h^k(x) \leftarrow \text{FIND\_CCE}(\bar{Q}_h^k, \underline{Q}_h^k, s)$ (cf. Algorithm 2 in Xie et al. (2020))
16:       $\bar{V}_h^k(s) \leftarrow \mathbb{E}_{(a,b) \sim \pi_h^k(s)} \bar{Q}_h^k(s, a, b)$ for each $s$
17:       $\underline{V}_h^k(s) \leftarrow \mathbb{E}_{(a,b) \sim \pi_h^k(s)} \underline{Q}_h^k(s, a, b)$ for each $s$
18:    **end for**
19: **else**
20:    $\bar{Q}_h^k \leftarrow \bar{Q}_h^{k-1}, \underline{Q}_h^k \leftarrow \underline{Q}_h^{k-1}, \pi_h^k \leftarrow \pi_h^{k-1}, \forall h \in [H]$
21: **end if**

---

*Proof of Theorem 3.* Let $\{k_1, \cdots, k_{N_{\text{batch}}}\}$ be the episodes where the learner updates the policies. Moreover, we define $b_k = \max\{k_i : i \in [N_{\text{batch}}], k_i \leq k\}$. Moreover, we denote the marginals of $\pi_h^k$ as $\mu_h^k$ and $\nu_h^k$.

**Part 1: Upper bound of the batch number.** By the updating rule, we know there exists one $h \in [H]$ such that $\det(\Lambda_h^{k_i}) > \eta \cdot \det(\Lambda_h^{k_{i-1}})$, which further implies that

$$\prod_{h=1}^{H} \det(\Lambda_h^{k_i}) > \eta \cdot \prod_{h=1}^{H} \det(\Lambda_h^{k_{i-1}}).$$

This yields that

$$\prod_{h=1}^{H} \det(\Lambda_h^{k_{N_{\text{batch}}}}) > \eta^{N_{\text{batch}}} \cdot \prod_{h=1}^{H} \det(\Lambda_h^0) = \eta^{N_{\text{batch}}} \cdot \lambda^{dH}. \tag{34}$$

On the other hand, for any $h \in [H]$, we have

$$\prod_{h=1}^{H} \det(\Lambda_h^{k_{N_{\text{batch}}}}) \leq \prod_{h=1}^{H} \det(\Lambda_h^{K+1}). \tag{35}$$

Furthermore, by the fact that $\|\phi(\cdot, \cdot)\|_2 \leq 1$, we have

$$\Lambda_h^{K+1} = \sum_{k=1}^{K} \phi(s_h^k, a_h^k, b_h^k) \phi(s_h^k, a_h^k, b_h^k)^\top + \lambda I_d \preceq (K + \lambda) \cdot I_d,$$

which implies that

$$\prod_{h=1}^{H} \det(\Lambda_h^{K+1}) \leq (K + \lambda)^{dH}. \tag{36}$$

Combining (34), (35), and (36), we obtain that

$$N_{\text{batch}} \leq \frac{dH}{\log \eta} \log \left(1 + \frac{K}{\lambda}\right),$$

which concludes the first part of the proof.

**Part2: Regret.** We need the following two lemmas to bound the regret.

**Lemma 11.** *For any $(s, a, b, k, h)$, it holds that*

$$\underline{Q}_h^k(s, a, b) - 2(H - h + 1)\epsilon \leq Q_h^{\mu^k, *}(s, a, b) \leq Q_h^{*, \nu^k}(s, a, b) \leq \bar{Q}_h(s, a, b) + 2(H - h + 1)\epsilon,$$

*and*

$$\underline{V}_h^k(s) - 2(H - h + 2)\epsilon \leq V_h^{\mu^k, *}(s) \leq V_h^{*, \nu^k}(s) \leq \bar{V}_h^k(s) + 2(H - h + 2)\epsilon.$$

*Proof.* See Lemma 5 of Xie et al. (2020) for a detailed proof. □

**Lemma 12.** *For all $(s, a, b, h, k)$ and any fixed policy pair $(\mu, \nu)$, it holds with probability at least $1 - \delta$ that*

$$\left| \left\langle \phi(s, a, b), \bar{w}_h^{b_k} \right\rangle - Q_h^{\mu, \nu}(s, a, b) - \mathbb{P}_h \left( \bar{V}_{h+1}^{b_k} - V_{h+1}^{\mu, \nu} \right)(s, a, b) \right| \leq \Gamma_h^{b_k}(s, a, b),$$

$$\left| \left\langle \phi(s, a, b), \underline{w}_h^{b_k} \right\rangle - Q_h^{\mu, \nu}(s, a, b) - \mathbb{P}_h \left( \underline{V}_{h+1}^{b_k} - V_{h+1}^{\mu, \nu} \right)(s, a, b) \right| \leq \Gamma_h^{b_k}(s, a, b).$$

*Proof.* See Lemma 3 of Xie et al. (2020) for a detailed proof. □

We define

$$\delta_h^k := \bar{V}_h^{b_k}(s_h^k) - \underline{V}_h^{b_k}(s_h^k),$$
$$\zeta_h^k := \mathbb{E}[\delta_{h+1}^k \mid s_h^k, a_h^k, b_h^k] - \delta_{h+1}^k,$$
$$\bar{\gamma}_h^k := \mathbb{E}_{(a,b) \sim \pi_h^k(s_h^k)}[\bar{Q}_h^{b_k}(s_h^k, a, b)] - \bar{Q}_h^{b_k}(s_h^k, a_h^k, b_h^k),$$
$$\underline{\gamma}_h^k := \mathbb{E}_{(a,b) \sim \pi_h^k(s_h^k)}[\bar{Q}_h^{b_k}(s_h^k, a, b)] - \bar{Q}_h^{b_k}(s_h^k, a_h^k, b_h^k).$$

By Lemma 11, we have

$$
\begin{aligned}
V_1^{*,\nu^k}(s_1) - V_1^{\pi^k,*}(s_1) &\le \bar{V}_1^k(s_1) - \underline{V}_1^k(s_1) + 8K\epsilon \\
&\le \bar{V}_1^k(s_1) - \underline{V}_1^k(s_1) + \frac{8}{K} \\
&= \bar{V}_1^{b_k}(s_1) - \underline{V}_1^{b_k}(s_1) + \frac{8}{K} \\
&= \delta_1^k + \frac{8}{K},
\end{aligned}
\tag{37}
$$

where the second inequality uses that $\epsilon = 1/(KH)$. By definition, we have

$$
\begin{aligned}
\delta_h^k &= \bar{V}_h^{b_k}(s_h^k) - \overline{V}_h^{b_k}(s_h^k) \\
&= \mathbb{E}_{(a,b)\sim\pi_h^k(s_h^k)}[\bar{Q}_h^k(s_h^k,a,b)] - \mathbb{E}_{(a,b)\sim\pi_h^k(s_h^k)}[\underline{Q}_h^{b_k}(s_h^k,a,b)] \\
&= \bar{Q}_h^k(s_h^k,a_h^k,b_h^k) - \underline{Q}_h^{b_k}(s_h^k,a_h^k,b_h^k) + \bar{\gamma}_h^k - \underline{\gamma}_h^k.
\end{aligned}
\tag{38}
$$

Meanwhile, for any $(s,a,b,k,h)$, we have

$$
\begin{aligned}
\bar{Q}_h^{b_k}(s,a,b) - \underline{Q}_h^{b_k}(s,a,b) &= [(\bar{w}_h^{b_k})^\top \phi(s,a,b) + \Gamma_h^{b_k}(s,a,b)] - [(\underline{w}_h^{b_k})^\top \phi(s,a,b) - \Gamma_h^{b_k}(s,a,b)] \\
&= (\bar{w}_h^{b_k} - \underline{w}_h^{b_k})^\top \phi(s,a,b) + 2\Gamma_h^{b_k}(s,a,b).
\end{aligned}
$$

By Lemma 12, we further have

$$
\bar{Q}_h^{b_k}(s,a,b) - \underline{Q}_h^{b_k}(s,a,b) \le \mathbb{P}_h(\bar{V}_{h+1}^{b_k} - \underline{V}_{h+1}^{b_k})(s,a,b) + 4\Gamma_h^{b_k}(s,a,b).
\tag{39}
$$

Plugging (39) into (38) gives that

$$
\begin{aligned}
\delta_h^k &\le \mathbb{P}_h(\bar{V}_{h+1}^{b_k} - \underline{V}_{h+1}^{b_k})(s,a,b) + 4\Gamma_h^{b_k}(s,a,b) + \bar{\gamma}_h^k - \underline{\gamma}_h^k \\
&= \delta_{h+1}^k + \zeta_h^k + \bar{\gamma}_h^k - \underline{\gamma}_h^k + 4\Gamma_h^{b_k}(s,a,b).
\end{aligned}
$$

Hence, we have

$$
\begin{aligned}
\sum_{k=1}^K \delta_1^k &\le \sum_{k=1}^K \sum_{h=1}^H (\zeta_h^k + \bar{\gamma}_h^k - \underline{\gamma}_h^k) + \sum_{k=1}^K \sum_{h=1}^H \Gamma_h^{b_k}(s_h^k,a_h^k,b_h^k) \\
&\le \tilde{O}(H \cdot \sqrt{HK}) + \sum_{k=1}^K \sum_{h=1}^H \Gamma_h^{b_k}(s_h^k,a_h^k,b_h^k),
\end{aligned}
\tag{40}
$$

where the last inequality uses the Azuma-Hoeffding inequality. Note that

$$
\frac{\Gamma_h^{b_k}(s_h^k,a_h^k,b_h^k)}{\Gamma_h^k(s_h^k,a_h^k,b_h^k)} \le \sqrt{\frac{\det(\Lambda_h^k)}{\det(\Lambda_h^{b_k})}} \le \sqrt{\eta},
$$

which implies that

$$
\sum_{k=1}^K \sum_{h=1}^H \Gamma_h^{b_k}(s_h^k,a_h^k,b_h^k) \le \sqrt{\eta} \sum_{k=1}^K \sum_{h=1}^H \Gamma_h^k(s_h^k,a_h^k,b_h^k) \le \tilde{O}(\sqrt{\eta d^3 H^4 K}),
\tag{41}
$$

where the last inequality uses the elliptical potential lemma (Abbasi-Yadkori et al., 2011). Combining (37), (40), and (41), together with the fact that $\eta = 2$, we obtain

$$
\text{Regret}(K) \le \tilde{O}(\sqrt{d^3 H^4 K}),
$$

which concludes the proof. $\qquad\square$

## D.3 Multi-batched V-learning

In this section, we introduce the multi-batched version of V-learning (Jin et al., 2021) for general-sum Markov games. Our goal is to minimize the following notion of regret. Let $V_{i,1}^\pi(s_1)$ be the expected cumulative reward that the $i^{\text{th}}$ player will receive if the game starts at initial state $s_1$ at the $1^{\text{st}}$ step and all players follow joint policy $\pi$. Let $\hat{\pi}^k$ be the policy executed at the $k^{\text{th}}$ episode. The regret is defined as

$$\text{Regret}(K) = \sum_{k=1}^K \max_j \left( V_{j,h}^{\dagger, \hat{\pi}_{-j,h}^k} - V_{j,h}^{\hat{\pi}_h^k} \right)(s_1), \tag{42}$$

where $V_{j,h}^{\dagger, \hat{\pi}_{-j,h}^k}$ is the best response of the $j^{\text{th}}$ player when other players follow $\hat{\pi}$.

---

**Algorithm 6** Multi-batched V-learning

---

1: **Initialize:** $V_h(s) \leftarrow H + 1 - h, N_h(s) \leftarrow 0, \pi_h(a \mid s) \leftarrow 1/A$, trigger set $\mathcal{L} \leftarrow \{2^{i-1} \mid 2^i \le KH, i = 1, 2, \ldots\}$
2: **for** episode $k = 1, \ldots, K$ **do**
3:     Receive $s_1$.
4:     **for** $h = 1, \ldots, H$ **do**
5:         take action $a_h \sim \pi_h^{\tilde{k}}(\cdot \mid s_h)$, observe $r_h$ and $s_{h+1}$.
6:         $t = N_h(s_h) \leftarrow N_h(s_h) + 1$.
7:         $\tilde{V}_h(s_h) \leftarrow (1 - \alpha_t)\tilde{V}_h(s_h) + \alpha_t(r_h + V_{h+1}(s_{h+1}) + \beta(t))$
8:         $V_h(s_h) \leftarrow \min\left\{H + 1 - h, \tilde{V}_h(s_h)\right\}$
9:         $\pi_h^k(\cdot \mid s_h) \leftarrow \text{ADV\_BANDIT\_UPDATE}\left(a_h, \frac{H - r_h - V_{h+1}(s_{h+1})}{H}\right)$ on $(s_h, h)^{\text{th}}$ adversarial bandit.
10:     **end for**
11:     **if** $\exists s, N_h(s) \in \mathcal{L}$ **then**
12:         $\tilde{k} \leftarrow k$
13:     **end if**
14: **end for**

---

The multi-batched V-learning algorithm maintains a value estimator $V_h(s)$, a counter $N_h(s)$, and a policy $\pi_h(\cdot \mid s)$ for each $s$ and $h$. We also maintain $S \times H$ different adversarial bandit algorithms. At step $h$ in episode $k$, the algorithm is divided into three steps: policy execution, $V$-value update, and policy update. In policy execution step, the algorithm takes action $a_h$ according to $\pi_h^{\tilde{k}}$, and observes reward $r_h$ and the next state $s_{h+1}$, and updates the counter $N_h(s_h)$. Note that $\pi_h^{\tilde{k}}$ is updated only when the visiting count of some state $N_h(s)$ doubles. Therefore it ensures a low batch number.

In the $V$-value update step, we update the estimated value function by

$$\tilde{V}_h(s_h) = (1 - \alpha_t)\tilde{V}_h(s_h) + \alpha_t(r_h + V_{h+1}(s_{h+1}) + \beta(t)),$$

where the learning rate is defined as

$$\alpha_t = \frac{H + 1}{H + t}, \quad \alpha_t^0 = \prod_{j=1}^t (1 - \alpha_j), \quad \alpha_t^i = \alpha_i \prod_{j=i+1}^t (1 - \alpha_j).$$

and $\beta(t)$ is the bonus to promote optimism.

In the policy update step, the algorithm feeds the action $a_h$ and its "loss" $\frac{H - r_h + V_{h+1}(s_{h+1})}{H}$ to the $(s_h, h)^{\text{th}}$ adversarial bandit algorithm. Then it receives the updated policy $\pi_h(\cdot \mid s_h)$.

**The ADV_BANDIT_UPDATE subroutine** Consider a multi-armed bandit problem with adversarial loss, where we denote the action set by $\mathcal{B}$ with $|\mathcal{B}| = B$. At round $t$, the learner determines a policy $\theta_t \in \Delta_{\mathcal{B}}$, and the adversary chooses a loss vector $\ell_t \in [0, 1]^B$. Then the learner takes

an action $b_t$ according to policy $\theta_t$, and receives a noisy bandit feedback $\tilde{\ell}_t(b_t) \in [0,1]$, where $\mathbb{E}\left[\tilde{\ell}_t(b_t) \mid \ell_t, b_t\right] = \ell_t(b_t)$. Then, the adversarial bandit algorithm performs updates based on $b_t$ and $\tilde{\ell}_t(b_t)$, and outputs the policy for the next round $\theta_{t+1}$.

We first state our requirement for the adversarial bandit algorithm used in V-learning, which is to have a high probability weighted external regret guarantee as follows.

**Assumption 4.** *For any $t \in \mathbb{N}$ and any $\delta \in (0,1)$, with probability at least $1 - \delta$, we have*

$$\max_{\theta \in \Delta_\mathcal{B}} \sum_{i=1}^{t} \alpha_t^i \left[\langle \theta_i, \ell_i \rangle - \langle \theta, \ell_i \rangle\right] \leq \xi(B, t, \log(1/\delta)). \tag{43}$$

*We further assume the existence of an upper bound $\Xi(B, t, \log(1/\delta)) \geq \sum_{t'=1}^{t} \xi\left(B, t', \log(1/\delta)\right)$ where (i) $\xi(B, t, \log(1/\delta))$ is non-decreasing in $B$ for any $t, \delta$; (ii) $\Xi(B, t, \log(1/\delta))$ is concave in $t$ for any $B, \delta$.*

As is shown in Jin et al. (2021), the Follow-the-Regularized-Leader (FTRL) algorithm satisfies the assumption with bounds $\xi(B, t, \log(1/\delta)) \leq O(\sqrt{HB\log(B/\delta)/t})$ and $\Xi(B, t, \log(1/\delta)) \leq O(\sqrt{HBt\log(B/\delta)})$. And we will use FTRL as the subroutine in our algorithm.

For two-player general sum Markov games, we let the two players run Algorithm 6 independently. Each player will use her own set of bonus that depends on the number of her actions. We have the following theorem for the regret of multi-batched V-learning.

**Theorem 13.** *Suppose we choose subroutine ADV_BANDIT_UPDATE as FTRL. For any $\delta \in (0,1)$ and $K \in \mathbb{N}$, let $\iota = \log(HSAK/\delta)$. Choose learning rate $\alpha_t$ and bonus $\{\beta(t)\}_{t=1}^{K}$ as $\beta(t) = c \cdot \sqrt{H^3 A \iota / t}$ so that $\sum_{i=1}^{t} \alpha_t^i \beta(i) = \Theta\left(\sqrt{H^3 A \iota / t}\right)$ for any $t \in [K]$, where $A = \max_i A_i$ Then, with probability at least $1 - \delta$, after running Algorithm for $K$ episodes, we have*

$$\text{Regret}(K) \leq O\left(\sqrt{H^5 S A \iota}\right). \tag{44}$$

*And the batch number is $O(HS\log K)$.*

*Proof.* The proof mainly follows Jin et al. (2021). Below we list some useful lemmas, which have already been proved in Jin et al. (2021).

**Lemma 14** (Lemma 10, Jin et al. (2021)). *The following properties hold for $\alpha_t^i$:*

*1. $\frac{1}{\sqrt{t}} \leq \sum_{i=1}^{t} \frac{\alpha_t^i}{\sqrt{i}} \leq \frac{2}{\sqrt{t}}$ and $\frac{1}{t} \leq \sum_{i=1}^{t} \frac{\alpha_t^i}{i} \leq \frac{2}{t}$ for every $t \geq 1$.*

*2. $\max_{i \in [t]} \alpha_t^i \leq \frac{2H}{t}$ and $\sum_{i=1}^{t} \left(\alpha_t^i\right)^2 \leq \frac{2H}{t}$ for every $t \geq 1$.*

*3. $\sum_{t=i}^{\infty} \alpha_t^i = 1 + \frac{1}{H}$ for every $i \geq 1$.*

The next lemma shows that $V_h^k$ is an optimistic estimation of $V_{j,h}^{\dagger, \hat{\pi}_{-j,h}^k}$

**Lemma 15** (Lemma 13, Jin et al. (2021)). *For any $\delta \in (0,1]$, with probability at least $1 - \delta$, for any $(s, h, k, j) \in \mathcal{S} \times [H] \times [K] \times [m]$, $V_{j,h}^k(s) \geq V_h^{\dagger, \nu^k}(s)$*

And we define the pessimistic pessimistic V-estimations $\underline{V}$ that are defined similarly as $\tilde{V}$ and $V$. Formally, let $t = N_h^k(s)$, and suppose $s$ was previously visited at episodes $k^1, \ldots, k^t < k$ at the $h$-th step. Then

$$\underline{V}_{j,h}^k(s) = \max\left\{0, \sum_{i=1}^{t} \alpha_t^i \left[r_{j,h}\left(s, \boldsymbol{a}_h^{k^i}\right) + \underline{V}_{j,h+1}^{k^i}\left(s_{h+1}^{k^i}\right) - \beta(i)\right]\right\}. \tag{45}$$

The next lemma shows $\underline{V}_{j,h}^k(s)$ is a lower bound of $V_{j,h}^{\hat{\pi}_h^k}(s)$.

**Lemma 16** (Lemma 14, Jin et al. (2021)). *For any $\delta \in (0, 1]$, with probability at least $1 - \delta$, the following holds for any $(s, h, k, j) \in \mathcal{S} \times [H] \times [K] \times [m]$ and any player $j$, $\underline{V}_{j,h}^k(s) \leq V_{j,h}^{\hat{\pi}_h^k}(s)$.*

It remains to bound the gap $\sum_{k=1}^K \max_j \left( V_{1,j}^k - \underline{V}_{1,j}^k \right) (s_1)$. For each player $j$ we define $\delta_{j,h}^k := V_{j,h}^k \left( s_h^k \right) - \underline{V}_{j,h}^k \left( s_h^k \right) \geq 0$. The non-negativity here is a simple consequence of the update rule and induction. We need to bound $\delta_h^k := \max_j \delta_{j,h}^k$. Let $n_h^k = N_h^k \left( s_h^k \right)$ and suppose $s_h^k$ was previously visited at episodes $k^1, \ldots, k^{n_h^k} < k$ at the $h$-th step. Now by the update rule,

$$
\begin{aligned}
\delta_{j,h}^k &= V_{j,h}^k \left( s_h^k \right) - \underline{V}_{j,h}^k \left( s_h^k \right) \\
&\leq \alpha_{n_h^k}^0 H + \sum_{i=1}^{n_h^k} \alpha_{n_h^k}^i \left[ \left( V_{j,h+1}^{k^i} - \underline{V}_{j,h+1}^{k^i} \right) \left( s_{h+1}^{k^i} \right) + 2\beta_j^{\bar{k}}(i) \right] \\
&\leq \alpha_{n_h^k}^0 H + \sum_{i=1}^{n_h^k} \alpha_{n_h^k}^i \left[ \left( V_{j,h+1}^{k^i} - \underline{V}_{j,h+1}^{k^i} \right) \left( s_{h+1}^{k^i} \right) + 4\beta_j^k(i) \right] \\
&= \alpha_{n_h^k}^0 H + \sum_{i=1}^{n_h^k} \alpha_{n_h^k}^i \delta_{j,h+1}^{k^i} + O \left( H\xi \left( A_j, n_h^k, \iota \right) + \sqrt{H^3 \iota / n_h^k} \right)
\end{aligned}
\tag{46}
$$

The third line holds because we use FTRL as the subroutine, $\beta_j(t) = c \cdot \sqrt{H^3 A_j \iota / t}$, and due to our doubling scheme we have $\beta(n_h^{\tilde{k}}) \leq 2\beta(n_h^k)$. And in the last step we have used $\sum_{i=1}^t \alpha_t^i \beta_{j,i} = \Theta \left( H\xi \left( A_j, t, \iota \right) + \sqrt{H^3 A \iota / t} \right)$.

The remaining step is the same as Jin et al. (2021). Summing the first two terms w.r.t. $k$,

$$
\sum_{k=1}^K \alpha_{n_h^k}^0 H = \sum_{k=1}^K H \mathbb{I} \left\{ n_h^k = 0 \right\} \leq SH,
\tag{47}
$$

$$
\sum_{k=1}^K \sum_{i=1}^{n_h^k} \alpha_{n_h^k}^i \delta_{h+1}^{k^i} \leq \sum_{k'=1}^K \delta_{h+1}^{k'} \sum_{i=n_h^{k'}+1}^{\infty} \alpha_i^{n_h^{k'}} \leq \left( 1 + \frac{1}{H} \right) \sum_{k=1}^K \delta_{h+1}^k.
\tag{48}
$$

Putting them together,

$$
\begin{aligned}
\sum_{k=1}^K \delta_h^k &= \sum_{k=1}^K \alpha_{n_h^k}^0 H + \sum_{k=1}^K \sum_{i=1}^{n_h^k} \alpha_{n_h^k}^i \delta_{h+1}^{k^i} + \sum_{k=1}^K O \left( H\xi \left( A, n_h^k, \iota \right) + \sqrt{\frac{H^3 \iota}{n_h^k}} \right) \\
&\leq HS + \left( 1 + \frac{1}{H} \right) \sum_{k=1}^K \delta_{h+1}^k + \sum_{k=1}^K O \left( H\xi \left( A, n_h^k, \iota \right) + \sqrt{\frac{H^3 \iota}{n_h^k}} \right).
\end{aligned}
\tag{49}
$$

Recursing this argument for $h \in [H]$ gives

$$
\sum_{k=1}^K \delta_1^k \leq eSH^2 + e \sum_{h=1}^H \sum_{k=1}^K O \left( H\xi \left( A, n_h^k, \iota \right) + \sqrt{\frac{H^3 \iota}{n_h^k}} \right).
\tag{50}
$$

By pigeonhole argument,

$$
\begin{aligned}
\sum_{k=1}^K \left( H\xi \left( A, n_h^k, \iota \right) + \sqrt{H^3 \iota / n_h^k} \right) &= O(1) \sum_s \sum_{n=1}^{N_h^K(s)} \left( H\xi(A, n, \iota) + \sqrt{\frac{H^3 \iota}{n}} \right) \\
&\leq O(1) \sum_s \left( H\Xi \left( A, N_h^K(s), \iota \right) + \sqrt{H^3 N_h^K(s) \iota} \right) \\
&\leq O \left( HS\Xi(A, K/S, \iota) + \sqrt{H^3 SK\iota} \right),
\end{aligned}
\tag{51}
$$

where in the last step we have used concavity. Finally take the sum w.r.t $h \in [H]$, and plugging $\Xi(B, t, \log(1/\delta)) \leq O(\sqrt{HBt \log(B/\delta)})$ we have

$$\text{Regret}(K) \leq O\left(\sqrt{H^5 S A \iota}\right). \tag{52}$$

$\square$

