# OpenReview forum: "A Reduction-based Framework for Sequential Decision Making with Delayed Feedback"
_NeurIPS.cc/2023/Conference — NeurIPS 2023 poster_

### Official Review · Reviewer_EWXd · 2023-07-04

**Soundness:** 3 good
**Presentation:** 2 fair
**Contribution:** 3 good
**Rating:** 6
**Confidence:** 3

**Summary:**

The paper studies the problem of sequential decision-making problems with delayed feedback. The authors propose a reduction from standard sequential decision-making algorithms to the following problem providing regret guarantees in many settings: linear bandits, tabular RL, linear and general function approximations RL, tabular and linear zero-sum Markov Games, tabular general-sum Markov Games.



**Strengths:**

- The authors improve previous results in the delayed feedback setting for multi-armed bandit, linear bandit, tabular MDP, tabular multi-player general sum MGs.
- The authors provide the first regret guarantees (in the delayed feedback setting) for linear MDPs, general-function approximation MDPs, and tabular and linear two-player zero-sum MGs.
- The strategy is simple:
   - single-agent: apply multi-batched algorithm for RL to the delayed feedback setting running some extra steps depending on the feedback delay $\tau_k$.
  - zero-sum tabular and general-sum tabular: using the doubling trick to update the policy.
  - zero-sum linear: update the policy when there exists $h \in [H]$ such that $\text{det}(\Lambda^k_h) > \eta \cdot \text{det}(\Lambda_h)$.

**Weaknesses:**

### Presentation

The presentation of the paper can be improved. The paper proposes a general strategy to convert multi-batch algorithms to the delayed feedback setting, changing (if I understood correctly) when the policy is updated. This is a simple and elegant transformation and it is not very clear from the presentation where the authors divide the transformations into three sections without providing this connection between them. I suggest providing an algorithm for the general setting which changes the update depending on the setting.
Moreover, there are some minor issues in the way the algorithms are presented:
- provide the input parameters in a clear way (e.g. algorithm 1 what is $l_m$?, algorithm 2 how can we compute $\tau_t$ since it is a random variable, algorithm 4 what is i?)
- provide output to the algorithms.


### Novelty

The simplicity of the strategy can be seen as a lack of novelty (due also to the previous point (see Presentation)). I suggest making more clear the challenges.

**Questions:**

See weaknesses.

Some minor questions:
- In algorithm 2, do we know $\tau_t$ or are we waiting until we do not receive the feedback, and then we can evaluate $\tau_t$?
- From a technical point of view what are the main challenges in converting regret bounds from the "classic" setting to the delayed feedback one?

**Limitations:**

The paper is mainly theoretical and the algorithms presented are not easily usable in practice.

---

> ### Author Rebuttal · Authors · 2023-08-09
>
> Thanks for your careful reading and valuable comments!
>
> Q1: Presentation of the framework
>
> A1: In fact, the framework we propose in Algorithm 2 is already for the general setting, which covers both single-agent and multi-agent, both tabular and linear. This framework can transform any multi-batched algorithm into an algorithm in the delayed setting, regardless of the specific type of batches. The “three sections” you mentioned are a strategy for designing multi-batched algorithms, and are not part of the main framework.
>
> Q2: Parameters in Algorithm 1
>
> A2: $\ell^m$ is the length of the policy sequence $\pi^m$ calculated by the algorithm. It is not an input parameter. For example, in the phase elimination algorithm for linear bandits in Lattimore and Szepesvári(2020), $\ell^m=T_m\approx 2^m$ (for the definition of $T_m$ see Appendix C).
>
> Q3:$\tau_t$ in Algorithm 2
>
> A3: In algorithm 2, we do not know $\tau_t$ in advance. In fact, we do not need to know $\tau_t$ or evaluate it during the whole algorithm. We only wait until we receive enough feedback so that the stopping condition in this batch is satisfied. In other words, the algorithm does not rely on knowledge of $\tau_t$. It is just a mathematical symbol for ease of presentation.
>
> Q4: Parameter $i$ in Algorithm 4
>
> A4: In algorithm 4, $i$ is used to state the trigger set $\mathcal{L}$. $\mathcal{L}$ contains all powers of 2, as long as they do not exceed $KH$.
>
> Q5: Outputs of the algorithms
>
> A5: We will clarify the output of the algorithms in our revision. Most of them are sequential decision making algorithms that output a sequence of policies. Thanks for your suggestion.
>
> Q6: Novelty and simplicity
>
> A6: Our framework is simple yet effective, as it can handle various settings, and using our framework we obtain many results that match or surpass previous results. We believe that simplicity and effectiveness are also a form of novelty.
> Besides, we also summarize our novel contribution as three points. Please see our **common rebuttal** for more explanation.
>
> Q7: From a technical point of view what are the main challenges in converting regret bounds from the "classic" setting to the delayed feedback one?
>
> A7: The key point is to decompose the regret bound into two parts: (i) the regret bound of multi-batched algorithms in the classic setting; and (ii) the additional regret incurred by the waiting time at the end of each batch. The first part follows previous results and the second one uses the property of multi-batched algorithms and stochastic delay. Our proof is simple but effective in the sense that we can obtain a unified result for decision making with delayed feedback. Moreover, when specialized to concrete problems, the obtained results match or even improve existing results.

---

> > ### Comment · Reviewer_EWXd · 2023-08-21
> >
> > I would like to thank the authors for their detailed replies. I continue to recommend the acceptance, hoping the authors will address the clarity issues.

---

### Official Review · Reviewer_p5oe · 2023-07-05

**Soundness:** 4 excellent
**Presentation:** 3 good
**Contribution:** 3 good
**Rating:** 7
**Confidence:** 3

**Summary:**

This submission studies the problem of learning from delayed feedback in various online learning settings including multi-armed bandits, linear bandits, linear MDPs, RL with function approximation, and various markov game settings (among others). Delays are assumed to be i.i.d. random variables. Additionally, they are sometimes assumed to be subexponential random variables.

The authors main result is a framework for sequential decision making with delayed feedback. Their framework makes use of "multi-batched" algorithms, which, as the name suggests, runs in batches. In each batch, a multi-batched algorithm outputs a sequence of policies to be used one-after-one (which are trained using data from previous batches), as well as a stopping criterion which decides when to stop the current batch. After a batch is run, the new data is incorporated with the existing data and is used to produce a new sequence of policies and stopping rule for the next round.

At a high level, the framework converts a multi-batched algorithm for the setting of interest (e.g. linear bandits) to one which can handle delayed feedback. The key idea is to run the policy given by the multi-batched algorithm for some extra time-steps in order to satisfy the stopping criteria. The authors are then able to bound the additional regret due to the delayed feedback by bounding the number of additional steps needed to satisfy the stopping criteria.

By instantiating their framework in the various settings listed above, they can obtain provable guarantees on the learner's regret. The rates obtained improve upon existing results in some settings (e.g. linear bandits and tabular MDPs), are slightly worse in others (e.g. multi-armed bandits), and are the first of their kind in other settings (e.g., RL with general function approximation).

**Strengths:**

The problem of delayed feedback in online learning settings is a well-motivated problem with many real-world applications. Furthermore, the authors are the first to obtain results for learning with delayed feedback in a variety of (important) settings such as RL with general function approximation, as well as improve upon existing results in others such as linear bandits.

The two main strengths of this paper are the simplicity/clarity of the proposed framework and the sheer number of results the authors are able to obtain using their framework. It is impressive that results for so many important settings are able to be obtained by applying one simple framework. Moreover, all that is required to use their framework is a multi-batched algorithm, which already exist in the literature for many online learning settings.

**Weaknesses:**

At times, the writing feels a bit rushed and as a result the submission may be a hard read for someone who is not an expert in RL theory. Specifically, I felt that the mutli-agent results were somewhat lacking in detail. (For example, why is the CCE the "right" solution concept, what happens when all players do/don't play a CCE policy, etc.) This is probably due to the fact that the authors have lots of results, which they wish to highlight in the main body of the paper. One suggestion would be to move either subsection 5.1, 5.2, or 5.3 to the Appendix, and use the extra space to provide more background/detail/hand-holding for the reader.

**Questions:**

I was a bit confused by the notation in line 9 of Algorithm 2. Shouldn't this read (both in words and mathematically) "Collect trajectory feedback in this batch that is observed **by** the end of the episode"?

Would your framework be able to obtain stronger results if, instead of Assumption 2, the delays are assumed to be subGaussian random variables?

A reference is missing to "Banker Online Mirror Descent: A Universal Approach for Delayed Online Bandit Learning" by Huang et al.

**Limitations:**

The authors have adequately addressed the limitations of their work.

---

> ### Author Rebuttal · Authors · 2023-08-09
>
> Thanks for your careful reading and valuable comments!
>
> Q1: Writing and readability
>
> A1: To comply with the page limit, we have moved the details about multi-agent Markov Games to the appendix. We will polish the contents to ensure clarity and coherence for the readers in the revision. We appreciate your valuable feedback.
>
> Q2: Why we consider CCE in general-sum Markov Games?
>
> A2: Both NE and CCE are common learning objectives in Markov games. However, finding the Nash equilibrium is computationally hard in general-sum Markov games. In this case, a coarse correlated equilibrium (CCE) is a more tractable notion of equilibrium that strictly generalizes NE. Unlike NE, a CCE can be computed efficiently in polynomial time.
>
> Q3: Notations in line 9 of Algorithm 2
>
> A3: You are right. We will correct this in our final version. We mean that in the $k$th episode, after executing the batch policy we collect all the feedback that has been delayed until the end of the episode.
>
> Q4: Stronger results using subgaussian assumption
>
> A4: You are right. If we use subgaussian assumption instead of the subexponential assumption, we can get a stronger result since we use different concentration inequality. In fact, subgaussian assumption is stronger than subexponential assumption, since any subgaussian random variable is also subexponential.
>
> Q5: Reference to Huang et al.
>
> A5: Thanks for your reminder. We will add reference to this paper in our revision.

---

> > ### Comment · Reviewer_p5oe · 2023-08-11
> >
> > Thanks for the reply. As a follow-up to Q2, a third choice of equilibrium that is natural to consider is a correlated equilibrium (CE). Could your framework be adapted to learn a CE instead?
> >
> > As a follow-up to Q4, how do you hypothesize your regret rate in Theorem 1 would change under a subGaussian delay assumption?

---

> > > ### Author Response · Authors · 2023-08-11
> > >
> > > Thanks for your reply!
> > >
> > > Follow-up to Q2: We can also extend our framework to learn a CE. The key is to devise an algorithm that can learn a CE with low batch numbers in the undelayed environment, and then integrate it with our framework to handle the delayed environment. For instance, in the tabular Markov Game setting, we can modify the CE-version of V-learning algorithm from (Jin et al., 2021) by using a similar idea as in our Algorithm 4, and obtain a multi-batched version of the algorithm for this setting.
> > >
> > > Follow up to Q4: If we assume that the delay is $\sigma$-subgaussian, the problem-dependent constant $C_\tau$ in (7) will be $\sqrt{2\sigma^2\log(3KH/2\delta)}$ instead.

---

### Official Review · Reviewer_nwcC · 2023-07-07

**Soundness:** 3 good
**Presentation:** 2 fair
**Contribution:** 2 fair
**Rating:** 4
**Confidence:** 4

**Summary:**

The paper presents a new framework for handling stochastic delayed feedback in general sequential decision-making problems, encompassing bandits, single-agent Markov decision processes (MDPs), and Markov games (MGs). The authors introduce a novel reduction-based framework that converts any multi-batched algorithm for sequential decision-making with instantaneous feedback into a sample-efficient algorithm capable of managing stochastic delays. They provide various examples, demonstrating the efficacy of their framework in different scenarios, and present several new results, improving existing findings for linear bandits, tabular RL, and tabular MGs.


**Strengths:**

The paper addresses a significant and practical issue in sequential decision-making: stochastic delayed feedback. It is well-motivated with practical examples like recommendation systems, robotics, and video streaming.
The paper presents a novel, reduction-based framework that shows versatility across multiple domains (bandits, single-agent MDPs, and MGs), exhibiting a comprehensive approach.
The proposed solution offers theoretical advancements by improving the regret bounds for linear bandits and tabular RL and provides the first theoretical guarantees in RL with function approximation and multi-agent RL settings.
The authors effectively integrate existing multi-batched algorithms into their framework, illustrating the generality of their approach.


**Weaknesses:**

It seems that the contribution is rather incremental. The idea that multi-batched algorithms can be used in delayed setting is not new. What appears to be new is some new theorems bounding the regret when using multi-batched algorithms for delayed settings in a variety of settings.
The authors could strengthen their paper with some simple experiments to add to the theory


**Questions:**

Can you elaborate more on the specific limitations of the current state-of-the-art methods that your framework is intended to overcome?
Can your framework be generalized to handle other types of delays, or is it specifically designed for stochastic delays?


**Limitations:**

Can we develop new algorithms that work even better than existing multi-batch algorithms?
The paper leaves open the question of whether their results are tight for MDP and MG with delayed feedback. Thus, there is room for future work to tighten these results.
The paper suggests that a multi-batched algorithm with a smaller batch number for tabular Markov games could be derived, which may provide more efficiency, but does not provide it in this paper.
It remains unclear if their findings and the proposed framework will translate effectively into practical scenarios. The authors have not provided any real-world evaluations or use-cases to support their claims.

---

> ### Author Rebuttal · Authors · 2023-08-09
>
> Thanks for your careful reading and valuable comments!
>
> Q1:Is the contribution rather incremental? Explain specific limitations of the current state-of-the-art methods.
>
> A1:We respectfully disagree with the reviewer’s argument that our contribution is rather incremental. Please see our **common response** for more explanation.
>
> Q2: Can our framework handle other types of delays?
>
> A2: Our framework is specifically designed for stochastic delays. As we mentioned in section 1.1, our framework achieves a better result than directly applying the reduction from adversarial delay setting to stochastic delay setting.
>
> Q3:Other limitations
>
> A3:For other limitations such as designing multi-batched algorithms with smaller batch numbers, we have discussed in our paper and we leave this as future work. One possible direction is to follow the idea of Zhang et al. (2022b) and adapt their algorithm to linear RL/Markov game setting. In terms of numerical results, we will consider adding some experiments in the future version. Thanks for your suggestion.

---

> > ### Comment · Reviewer_nwcC · 2023-08-14
> > **Response to Authors**
> >
> > Thanks for clarifying the contributions. After reading the rebuttal and the other reviews I wouldn't be opposed to voting for acceptance if the authors can include additional discussion about the contribution in the camera-ready paper.

---

> > > ### Author Response · Authors · 2023-08-14
> > >
> > > Thank you for dedicating your time and providing your valuable support. We **promise** that we will incorporate the discussions into the revision. We would appreciate it if you would reconsider your score in light of our clarification.

---

### Official Review · Reviewer_bZKJ · 2023-07-21

**Soundness:** 3 good
**Presentation:** 2 fair
**Contribution:** 3 good
**Rating:** 6
**Confidence:** 4

**Summary:**

This paper provides a general framework for analysing any ‘mini-batch’ or rarely switching algorithm in the presence of delayed feedback. This approach covers bandits, finite horizon MDPs and finite horizon Markov games. The results provided match or improve on the best-known results for delayed feedback in settings that have been studied before, and provide the first bounds in some settings where delayed feedback has not been studied before.

**Strengths:**

It is nice to have a comprehensive study of delayed feedback in various settings. It is interesting that phase-based algorithms allow one to deal with delayed feedback effectively in a variety of settings. It is nice that the general framework provided in this paper allows us to recover the same bounds as for algorithms for specific settings and provides new bounds in settings where delayed feedback has not been studied previously.

**Weaknesses:**

The writing and clarity could be improved. I also found the structure quite confusing with a couple of pages dedicated to Markov games while a lot of the content needed to understand that section (e.g. related work and algorithms) relegated to the appendix. The amount of space dedicated to Markov games also meant that there was not sufficient room to discuss the results for delayed feedback in bandits/RL, nor for all details of the method/results to be provided (see the many questions below).
The idea of using phase-based algorithms which switch arms less frequently for delayed feedback is not new and has been used in e.g. Lancewicki et al (2021), Howson et al (2021), Pike-Burke et al (2018), Vakili et al (2023),… This should be stated in section 4. Additionally, I believe Vakili et al (2023) also provide the same bound for the linear bandit setting with delayed feedback as is established in this paper, so this should also be checked and added.

References: Vakili, Sattar, et al. "Delayed Feedback in Kernel Bandits." ICML, 2023.
Pike-Burke, Ciara, et al. "Bandits with delayed, aggregated anonymous feedback." ICML, 2018


**Questions:**

The two results in e.g. theorem 1 are a bit confusing and the difference between them is not well explained. Is the second one (7) on the expected regret and the first (6) on the high probability regret? For (6), I am assuming that q is a parameter which can be chosen, can any guidance be provided for how to choose it? Does it depend on \delta?
I am interested in whether experimentally these rarely switching algorithms actually perform better than algorithms tailored to the specific setting, or just versions of optimistic algorithms that only use available data?
A couple of times ‘minimax optimal’/’tight’ is stated. Are there lower bounds for all these settings with delayed feedback?
Can the lower bounds/best results for the non-delayed settings be added to the table of results to enable us to clearly understand what the impact of delayed feedback is?
In the general algorithm provided in section 4 and algorithm 2, do we need to know the delay distribution to define the batch lengths? The example in the text would suggest that we do, but the pseudo-code does not mention it.
In the introduction it is mentioned that the sub-exponential assumption on the delay distribution is only sometimes needed, yet I cannot find any discussion of when it is used for the main results. Is it needed for all results in Theorem 1?


**Limitations:**

Very brief discussion of limitations in conclusion

---

> ### Author Rebuttal · Authors · 2023-08-09
>
> Thanks for your careful reading and valuable suggestions!
>
> Q1. Writing and clarity
>
> A1. To comply with the page limit, we have moved the details about Markov Games to the appendix. We will arrange the contents to ensure clarity and coherence for the readers. We appreciate your valuable feedback.
>
> Q2. Novelty of our idea
>
> A2. We point out that unlike previous works that propose different phase-based algorithms and analyze them case by case, our work introduces a generic class of algorithms that can be integrated with any multi-batched algorithm in a black-box manner. We also provide a unified theoretical analysis for the proposed generic algorithm. Moreover, papers such as Howson et al (2021) and Pike-Burke et al (2018) fail to fully exploit the potential of multi-batched algorithms, and their bounds are inferior to ours. (see Table 1 for a comparison of the results)
>
> Q3. Checking Vakili et al (2023)
>
> A3. Our work does not consider the kernel setting, but we conjecture that our framework can handle this problem by designing a multi-batched algorithm for kernel bandits. We appreciate your suggestion of this related work and we will discuss it further in the revision.
>
> Q4.Explanation on results in Theorem 1
>
> A4. Both (6) and (7) are high-probability regret bounds. Here (7) holds with probability $1 - \delta$. Thank you for pointing this out and we will clarify in the revision. The main difference is the use of concentration inequality for the number of delays. (6) uses concentration for sub-exponential delays, and (7) uses concentration in the quantile form. The parameter $q$ can be any real number in (0,1) and does not rely on other parameters such as $\delta$.
>
> Q5. Numerical experiments
>
> A5.We will consider adding numerical experiments to compare the performance with algorithms tailored to the specific setting. However, we note that all these algorithms are optimistic algorithms, and the main difference lies in the timing of policy update, which directly affects the regret.
>
> Q6.Lower bounds
>
> A6.Here we can give a simple lower bound. Consider the case when the delay $\tau$ is a constant. For this case a trivial lower bound is $\Omega(R\tau)$ where $R$ is the upper bound of reward in each episode ($R=1$ in bandits, and $R=H$ in episodic MDP), since if we run the algorithm for only $\tau$ episodes, then the algorithm cannot observe anything, and it cannot do better than tossing a coin. Together with existing lower bound for undelayed problems, we obtain a lower bound $\Omega(\text{undelayed lower bound} + R\tau)$. By this lower bound, we know our results are tight in bandits setting. In MDPs and MGs, we improve existing results or provide the first line of study.
>
> Q7.Do we need to know the delay distribution in advance?
>
> A7.We do not need to know the delay distribution in advance. Moreover, we do not have to predefine the length of each batch beforehand. We simply run the batched policy until we collect enough data for the batch to stop.
>
> Q8.Where is the sub-exponential assumption used?
>
> A8.The sub-exponential assumption on the delay distribution is needed when we derive regret bound (7) in Theorem 1. It is used for concentration of delays.

---

> > ### Comment · Reviewer_bZKJ · 2023-08-11
> >
> > Thanks to the authors for answering some of my questions. However I think there are still some things that are not clear.
> >
> > Q2: I agree that the unifying framework the authors provide is new. However, I think that it is good scientific practice to acknowledge that the general idea of using rarely switching algorithms to mitigate the effects of the delayed feedback is not new, so it would be good if the authors mentioned that in Section 4.
> >
> > Q3: Vakili et al (2023) already uses a rarely switching algorithm since it is based on the BPE algorithm of Li & Scarlett (2022). Does the BPE algorithm thus fit into the framework in this paper? In which case what results are obtained? Also note that when the results of Vakili et al (2023) are applied to the linear kernel setting, they improve upon those of Howson et al (2021) to obtain the same $E[\tau]$ penalty as is obtained in this work. Therefore, the table on pg2 should be updated to include the results of Vakili et al (2023) for linear bandits.
> >
> > Q8: Does this mean that the sub-exponential assumption is not used for the regret bound in (6)? This should be mentioned.
> >
> >
> > Li, Z. and Scarlett, J., 2022. Gaussian process bandit optimization with few batches. In International Conference on Artificial Intelligence and Statistics (pp. 92-107).

---

> > > ### Author Response · Authors · 2023-08-11
> > >
> > > Thanks for your reply!
> > >
> > > Q2: We will add discussions on the ideas of previous works and emphasize our contributions in the revision. Thank you for your suggestion.
> > >
> > > Q3: In the kernel bandit setting, the BPE algorithm also fits into our framework. In this case, our framework gives a $\tilde{O}(\Lambda\sqrt{T\gamma_T}+\mathbb{E}[\tau]+C_{\tau})$ regret upper bound when the delays are subexponential (For the notation $\Lambda$ and $\gamma_T$ see Li & Scarlett (2022)). We will modify Table 1 to include results of Vakili et al (2023). Thank you for your suggestion.
> > >
> > > Q8: You are right. The sub-exponential assumption is not used for the regret bound in (6). We will make this more clear in our revision.

---

### Author Rebuttal · Authors · 2023-08-09

## Common rebuttal for the contributions of our paper:

We respectfully disagree with the reviewers’ argument that our contribution is rather incremental. Here we describe some drawbacks of previous works, and how our work addresses them.

The previous works suffer from several drawbacks: (i) they require case by case algorithm design and analysis; (ii) their regret bound is not tight; (iii) they do not explore some settings (linear MDPs and MGs).

Our work addresses these challenges, and our contributions are as follows:

(1)new framework: We propose **a generic class of algorithms** that can be integrated with **any** multi-batched algorithm in a black-box fashion. Meanwhile, we provide a **unified** theoretical analysis for the proposed generic algorithm.

(2)improved results and new results: By applying our framework to different settings, we obtain state-of-the-art regret bounds for multi-armed bandits, and derive sharper results for linear bandits and tabular RL, which significantly improve existing results.

(3)new algorithm: we design new multi-batched algorithms for multi-agent Markov games, and handle the delayed feedback in Markov games by combining our framework and the newly designed algorithm.

---

### Decision · Program_Chairs · 2023-09-21

**Decision:**

Accept (poster)

**Comment:**

This paper considers a general setting for sequential decision making, subsuming reinforcement learning with function approximation and multi-agent reinforcement learning, and considers the problem of learning under delayed feedback. The main result is a reduction-based framework which lifts any multi-batched algorithm for sequential decision making into an algorithm capable of handling delayed feedback. Using this reduction, the derive new results for a number of settings including reinforcement learning with linear MDPs or bounded eluder dimension.

Reviewers felt that the paper's approach was novel and potentially impactful. In particular, the authors derive the first and only existing results for learning with delayed feedback in a number of single- and multi-agent RL frameworks. While phased-based algorithms have been used in a number of prior works with delayed feedback, the significance of this paper is to provide a unified, black-box approach, and to derive a bounds for a comprehensive set of settings as a consequence. As such, there was a consensus after the discussion period that the paper should be accepted.

For the final version of the paper, the authors are encouraged to incorporate the clarifications, improvements to writing, and references to related work suggested by the reviewers.